# Latent-EnSF: A Latent Ensemble Score Filter for High-Dimensional Data Assimilation with Sparse Observation Data

**Phillip Si, Peng Chen**
Department of Computational Sciences and Engineering, Georgia Institute of Technology
{psi6, pchen402}@gatech.edu

## Abstract

Accurate modeling and prediction of complex physical systems often rely on data assimilation techniques to correct errors inherent in model simulations. Traditional methods like the Ensemble Kalman Filter (EnKF) and its variants as well as the recently developed Ensemble Score Filters (EnSF) face significant challenges when dealing with high-dimensional and nonlinear Bayesian filtering problems with sparse observations, which are ubiquitous in real-world applications. In this paper, we propose a novel data assimilation method, Latent-EnSF, which leverages EnSF with efficient and consistent latent representations of the full states and sparse observations to address the joint challenges of high dimensionality in states and high sparsity in observations for nonlinear Bayesian filtering. We introduce a coupled Variational Autoencoder (VAE) with two encoders to encode the full states and sparse observations in a consistent way guaranteed by a latent distribution matching and regularization as well as a consistent state reconstruction. With comparison to several methods, we demonstrate the higher accuracy, faster convergence, and higher efficiency of Latent-EnSF for two challenging applications with complex models in shallow water wave propagation and medium-range weather forecasting, for highly sparse observations in both space and time.

## 1 Introduction

Many complex physical systems are traditionally modeled by partial differential equations (PDEs). First-principle PDE-based modeling and simulation have proven to be powerful in predicting complex systems across various scientific and engineering fields. However, in practical applications, significant discrepancies between simulation-based predictions and reality may arise from sources such as model inadequacy, uncertainties in model parameters, boundary and/or initial conditions, external forcing/loading terms, numerical approximation errors, and more. Meanwhile, data-driven machine learning (ML) models like FourCastNet (Pathak et al., 2022) have been significantly developed in recent years, demonstrating extreme efficiency for system predictions as in weather forecasting. The computational efficiency also makes these ML models particularly well-suited for ensemble forecasting and uncertainty quantification. However, these models are often limited by a divergence from reality over the long run, resulting in significant accumulated errors in long-term predictions due to various uncertainties and the autoregressive structure of the ML models.

To mitigate discrepancies and accumulated errors and achieve more accurate predictions, data assimilation plays an essential role by incorporating additional observational data into current PDE/ML-based prediction models. By adjusting the PDE/ML state to align with the observational data, the accuracy of future state predictions can be markedly improved. Data assimilation (Sanz-Alonso et al., 2023; Asch et al., 2016) approaches such as the Kalman filter (KF) (Evensen, 1994), particle filters (Künsch, 2013), and their variants have been widely applied to problems including weather prediction, geophysical modeling, robotics, and many other areas.

The basis of many current data assimilation methods used in practice is the KF (Evensen, 1994). It parameterizes its state with the mean and covariance, and provides the optimal solution to the data assimilation problem when certain linearity properties of the transition and observation functions

are met. However, in practice, KFs become computationally expensive for high-dimensional states and can be biased for nonlinear problems. An alternative is the particle filter, which represents the density by an ensemble of particles. This approach relaxes the assumption of a specific distributional form, such as a Gaussian, that is inherent in distribution-based approaches like the KF. The EnKF (Kalman, 1960) is a particular particle filter that originates from the KF; instead of updating the covariance matrix, it computes the sample covariance of the ensemble of particles. Due to its robust performance, it has become widely used in applications today. However, most particle filters, including EnKF, suffer from the curse of dimensionality by requiring an exponentially large number of particles to accurately describe the distribution in high dimensions. To address this challenge, various extensions to EnKF have been proposed, such as the Localized Ensemble Transform Kalman Filter (LETKF) (Hunt et al., 2007), which applies covariance localization techniques along with advancements from the Ensemble Transform Kalman Filter (Bishop et al., 2001).

Nonetheless, several major roadblocks hinder efficient data assimilation for high-dimensional systems. Standard data assimilation approaches have limited capabilities in these environments, often relying on assumptions such as localization to remain applicable. High-dimensional systems and complex models can also constrain the capability and efficiency of methods like 4D-Var (Rabier & Liu, 2003), which requires repeatedly propagating through the physical model. Recently, DiffDA (Huang et al., 2024) employs diffusion models and masked interpolations of weather data to assimilate sparse observations, though the approach falls behind pure interpolation after a few steps. Meanwhile, Rozet & Louppe (2024) utilizes conditional score-based generative models to sample from the trajectory space conditioned on observations; however, this addresses a fundamentally different problem–sampling an entire trajectory conditioned on observations rather than performing the real-time filtering needed for practical applications. Bao et al. (2024b) proposed a novel method called Ensemble Score Filter (EnSF), which has achieved great results in high-dimensional data assimilation by leveraging score-based diffusion models to efficiently sample from the posterior distribution. Nonetheless, sparse observations remain very challenging for EnSF, limiting its applicability in real-world scenarios where observational data are scarce. Similarly, ROAD (Chen et al., 2023) develops latent dynamics through a decoder but relies on dense observations.

In this work, we propose an efficient data assimilation method called Latent-EnSF (shown in Figure 1) that builds on Ensemble Score Filters (EnSF) (Bao et al., 2024b) to address the joint challenges of high dimensionality and data sparsity. We construct coupled Variational Autoencoders (VAE) (Kingma & Welling, 2014) to encode the sparse observations into a latent space, where we perform data assimilation using EnSF with diffusion-based sampling. Importantly, our approach addresses the key weakness of EnSF when dealing with sparse observations while retaining its ability to efficiently assimilate observations for high-dimensional nonlinear systems. We demonstrate this performance for both PDE and ML-based prediction models on challenging examples, experimenting with data that is sparse in both space and time.

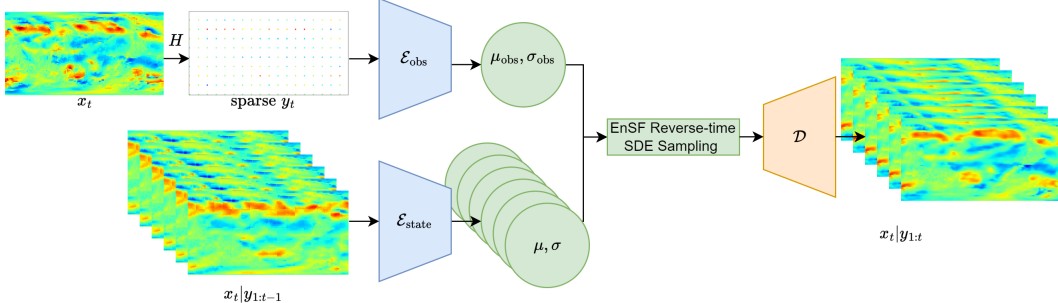

Figure 1: Flow of the Latent-EnSF. An ensemble of prior states $x_t|y_{1:t-1}$ is assimilated with sparse observation $y_t$ in the latent autoencoder space to obtain samples $x_t|y_{1:t}$ from the posterior.

We will begin by formally introducing data assimilation problems along with a review of the EnSF in Section 2, then explain how we integrate this with coupled variational autoencoders in Section 3 to form our Latent-EnSF, and finally conclude with experiments in Section 4.

## 2   DYNAMICAL SYSTEMS AND DATA ASSIMILATION

To model a dynamical physical system, we denote the state at a certain time $t$ as $x_t \in \mathbb{R}^d$, where $d$ is typically very high for complex models. For a system with discrete time steps $t = 1, 2, \ldots$, given initial condition $x_0$, we then model the evolution of the state from the time $t - 1$ to time $t$ as

$$x_t = M(x_{t-1}, \varepsilon_t), \tag{1}$$

where $\varepsilon_t$ is a process noise term coming from a known distribution; this term accounts for hidden interactions and numerical errors. The corresponding observation $y_t \in \mathbb{R}^m$ at time $t$ is given by

$$y_t = H(x_t) + \gamma_t, \tag{2}$$

where $\gamma_t$ (also coming from a known distribution, typically Gaussian) represents the observation noise because of instrumental inaccuracies, etc. Both the model map $M : \mathbb{R}^d \times \mathbb{R}^d \to \mathbb{R}^d$ and the observation map $H : \mathbb{R}^d \to \mathbb{R}^m$ can be nonlinear.

The dynamical system model in Equation 1 may yield progressively inaccurate representations over time due to factors such as model inadequacy or input uncertainty. The goal of data assimilation is to integrate the observation data into the dynamical system model to recover an accurate representation of the true state. In practical applications, the observation data can be sparsely distributed in both space and time, making the recovery of the true state particularly challenging, especially for high-dimensional problems with a state dimension $d \gg 1$, and the observation dimension $m \ll d$.

### 2.1   BAYESIAN FILTERING

From a Bayesian filtering perspective (Dore et al., 2009), a data assimilation problem at each time $t$ can be divided into two steps: a prediction step that advances the dynamical system and an update step that assimilates the data. Assuming that the posterior density of the state $x_{t-1}$ given observation data $y_{1:t-1} = (y_1, \ldots, y_{t-1})$, denoted as $P(x_{t-1}|y_{1:t-1})$, is available at time $t - 1$, with $P(x_0|y_{1:0}) = P(x_0)$ given for time $t - 1 = 0$, then the prediction step provides the density of $x_t$ as

$$\textbf{Prediction: } P(x_t|y_{1:t-1}) = \int P(x_t|x_{t-1})P(x_{t-1}|y_{1:t-1})dx_{t-1}, \tag{3}$$

where $P(x_t|x_{t-1})$ represents the transition probability governed by the dynamical system in Equation 1. Here, $x_t$ depends on $y_{1:t-1}$ only through $x_{t-1}$. Let $P(y_t|x_t)$ denote the likelihood function of the data $y_t$ given state $x_t$ from Equation 2. Then, the update step provides the updated posterior density $P(x_t|y_{1:t})$ of the state $x_t$ given the new observation data $y_t$ by Bayes' rule as

$$\textbf{Update: } P(x_t|y_{1:t}) = \frac{P(y_t|x_t)P(x_t|y_{1:t-1})}{P(y_t|y_{1:t-1})}, \tag{4}$$

where the normalization constant or model evidence term $P(y_t|y_{1:t-1})$ is given by $P(y_t|y_{1:t-1}) = \int P(y_t|x_t)P(x_t|y_{1:t-1})dx_t$, which is typically intractable to compute.

### 2.2   DIFFUSION MODELS AND THE ENSEMBLE SCORE FILTER

A recent development for a new particle filter is the EnSF, where the ensemble of samples can be drawn from the posterior distribution through a diffusion process. It builds on the recent advancement in score-based generative modeling (Ho et al., 2020) in generating high-fidelity samples from a distribution by first computing a score function $\nabla_x P(x)$ and then sampling with it by solving stochastic differential equations (SDE) (Song et al., 2021). This circumvents knowing the true density as it only needs to generate samples from the posterior distribution. Notably, the diffusion model takes the form of a noisy forward stochastic differential equation process

$$dx = f(x, \tau)d\tau + g(\tau)dw, \tag{5}$$

which transforms data from an arbitrary distribution to an isotropic Gaussian for $\tau \in \mathcal{T} = [0, T]$. Here, $f(x, \tau)$ is a drift term, $g(\tau)$ is a diffusion term, and $w$ is a $d$-dimensional Wiener process. It has the corresponding reverse-time SDE that runs backward in time from $\tau = T$ to $0$ as

$$dx = [f(x, \tau) - g^2(\tau)\nabla_x \log P_\tau(x)]d\tau + g(\tau)d\bar{w}, \tag{6}$$

where $\bar{w}$ is another Wiener process independent of $w$. This reverse-time SDE can be solved to sample from the distribution $P(x)$ if the score function $\nabla_x \log P(x_\tau)$ is known, e.g., by the Euler–Maruyama scheme at the discrete time steps $0 = \tau_0 < \tau_1 < ... < \tau_k = T$.

Let $x_{t,\tau}$ denote the state at physical time $t$ and diffusion time $\tau$. Bao et al. (2024b) propose the posterior score of the update step in Equation 4 as

$$\nabla_x \log P(x_{t,\tau}|y_{1:t}) = \nabla_x \log P(x_{t,\tau}|y_{1:t-1}) + h(\tau)\nabla_x \log P(y_t|x_{t,\tau}), \quad (7)$$

with the prior score function in the first term given as an integral, see Appendix A, that is computed by Monte Carlo estimation, the likelihood function $P(y_t|x_{t,\tau})$ in the second term given explicitly from the observation map in Equation 2, and a monotonically decreasing damping function $h(\tau)$ with $h(0) = 1$ and $h(1) = 0$, e.g., $h(\tau) = 1 - \tau$. Note that the posterior score function is consistent with Bayes' rule in Equation 4 at $\tau = 0$. To this end, one step of the EnSF algorithm is presented in Algorithm 2 in Appendix A, along with the details on the prior and posterior score functions.

EnSF offers several advantages over EnKF, particularly in high dimensions, where EnKF typically requires many more ensemble members to accurately estimate the density, while EnSF leverages the explicit likelihood function and diffusion process to generate samples. Moreover, EnSF does not assume the approximate linearity of the dynamical system, making it highly applicable to high-dimensional and nonlinear filtering problems, such as a million-dimensional Lorenz-96 system (Bao et al., 2024b) and a surface quasigeostrophic model (Bao et al., 2024a).

## 2.3 CHALLENGE OF ENSF FOR SPARSE OBSERVATIONS

A key limitation of EnSF in high-dimensional nonlinear filtering problems occurs when the observation data are sparse, which is the case for most practical applications. For example, suppose that the observation map $H(x_t) = x_t[S]$, which only makes observations of $x_t$ in the dimensions corresponding to a small subset $S \subset \{1, ..., d\}$ with $|S| \ll d$. In this case, the gradient of the log-likelihood $\nabla_x \log P(y_t|x_t)$ vanishes in the dimensions outside of $S$, as shown in Figure 2 for pointwise observations of shallow water wave height with $|S| = d/100$. As the likelihood function in Equation 7 does

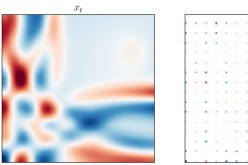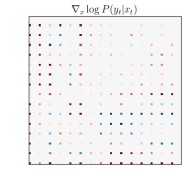

Figure 2: The gradient of the log-likelihood function $\nabla_x \log P(y_t|x_t)$ (right) vanishes at the points where the sparse observation data $y_t$ (middle) do not have any information of the state $x_t$ (left).

not account for the spatial correlation, the information from the sparse observations cannot be adequately assimilated to the unobserved states. The vanishing gradients significantly limit the effectiveness of the standard EnSF, which we demonstrate empirically in Section 4.1.1.

## 3 LATENT ENSEMBLE SCORE FILTER

To address the key limitation of EnSF in the case of sparse observations due to the vanishing gradient of the log-likelihood function, we propose a latent representation of the sparse observations by a variational autoencoder (VAE) and match this latent representation to the encoded full state for the sake of consistent data assimilation in the latent space, for which we employ EnSF with the additional advantage of VAE regularization of the latent variables.

The advantages are twofold: it enables us not only to exploit the benefits of standard latent score-based generative models (Vahdat et al., 2021; Rombach et al., 2022), which offer optimized sampling speed and expressivity, but also to create an expressive mapping from the sparse observation space to the latent space, with the output being a full-dimensional observation in the latent space that circumvents the vanishing gradient issue of the log-likelihood function. Additionally, the successful application of latent-diffusion models to videos (Blattmann et al., 2023) presents a potential avenue for exploring latent-assimilation coupled with latent dynamics.

## 3.1 Coupled Variational Autoencoders with Latent-Space Matching

The Variational Autoencoder (VAE) (Kingma & Welling, 2014), as well as its variants such as VQ-VAE (Van Den Oord et al., 2017) and InfoVAE (Zhao et al., 2019), provides a compressed representation of a high-dimensional distribution through bottlenecking. We employ VAEs to compress both the sparse observations and the full states into a consistent latent representation.

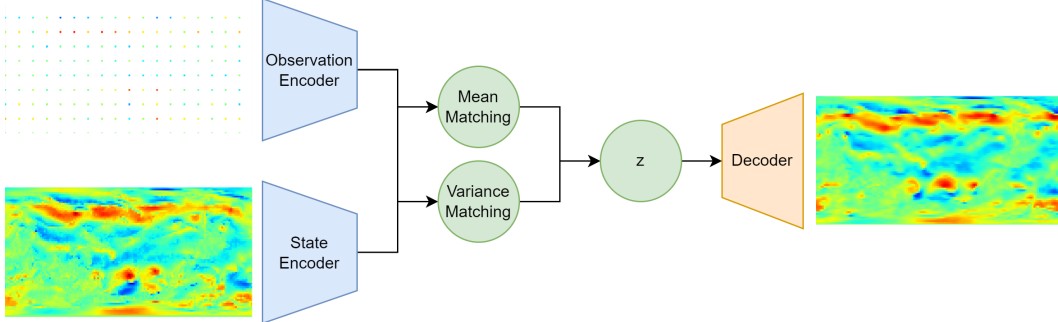

Figure 3: A coupled VAE for consistent latent representations of sparse observations and full states.

Specifically, as shown in Figure 3, we formulate a coupled VAE with two encoders. The state encoder, $\mathcal{E} : \mathbb{R}^d \to \mathbb{R}^{2r}$, encodes the full state $x_t$ into a latent variable $z_t$ with distribution $\nu_t = N(\mu_t, \text{diag}(\sigma_t^2))$, where $(\mu_t, \sigma_t^2) = \mathcal{E}(x_t)$ and $z_t = \mu_t + \sigma_t \cdot \epsilon_t$. The observation encoder, $\mathcal{E}_{\text{obs}} : \mathbb{R}^m \to \mathbb{R}^{2r}$, encodes the sparse observation $y_t$ into another latent variable $z_t^{\text{obs}}$ with distribution $\nu_t^{\text{obs}} = N(\mu_t^{\text{obs}}, \text{diag}((\sigma_t^{\text{obs}})^2))$, where $(\mu_t^{\text{obs}}, (\sigma_t^{\text{obs}})^2) = \mathcal{E}_{\text{obs}}(y_t)$ and $z_t^{\text{obs}} = \mu_t^{\text{obs}} + \sigma_t^{\text{obs}} \cdot \epsilon_t$. For a consistent reconstruction of the full state from the latent representation, we use the same decoder $\mathcal{D} : \mathbb{R}^r \to \mathbb{R}^d$. To assimilate the encoded latent data to the encoded latent state, we seek to match the two latent representations of the consistent input. In addition, we include a regularization of the latent distribution. To this end, we formulate the following loss function as

$$
\begin{aligned}
\ell_t(\theta) = & \|x_t - \mathcal{D}(z_t)\|_2^2 + \|x_t - \mathcal{D}(z_t^{\text{obs}})\|_2^2 && \text{(Reconstruction Term)} && (8) \\
& + \|\mathcal{E}(x_t) - \mathcal{E}_{\text{obs}}(y_t)\|_2^2 && \text{(Latent Matching Term)} && (9) \\
& + \lambda D_{\text{KL}}(\nu_t|\nu) + \lambda D_{\text{KL}}(\nu_t^{\text{obs}}|\nu) && \text{(Regularization Term)} && (10)
\end{aligned}
$$

where $\theta$ represents the parameters of the encoders $\mathcal{E}$ and $\mathcal{E}_{\text{obs}}$ and the decoder $\mathcal{D}$, for which we use convolutional neural networks (CNNs) in this work. In practice, we found that a latent matching for the mean is sufficient. The regularization by the Kullback-Leibler divergence between the latent distribution and the reference $\nu = N(0, I_r)$ of multivariate standard normal distribution

$$
D_{\text{KL}}(\nu_t|\nu) = \frac{1}{2} \sum_{i=1}^{r} \left( (\sigma_t)_i^2 + (\mu_t)_i^2 - \log((\sigma_t)_i^2) - 1 \right). \tag{11}
$$

$\lambda$ in Equation 10 is a weight parameter. We set $\lambda = 10^{-5}$ as in (Rombach et al., 2022) for the encoder of the latent diffusion model. We minimize an empirical loss function as a sum of the above loss function over all time steps $t$ in each of the training trajectories of the dynamical system.

Both latent representations use the same decoder to reconstruct the full state. By training a coupled VAE that minimizes the reconstruction error of the true state conditioned on the observations, we aim to obtain a latent representation of the sparse observations that is consistent with that of the full states. This approach is similar to the Generalized Latent Assimilation (GLA) representation proposed by Cheng et al. (2023) for EnKF, but with the added matching of the latent representations of the states and sparse observations, as illustrated in Figure 3. Experimentally, training the encoders without matching the latent representations can lead to a solution in which the latent observations are segregated from the latent states, rendering the data assimilation inaccurate. Using these two separate encoders, we can align the latent representations so that we are able to apply EnSF in a full-dimensional manner, even with extremely sparse observations.

### 3.2 LATENT ENSEMBLE SCORE FILTER

For simplicity, let the latent state $\xi_t$ and latent data $\zeta_t$ be defined as the encoded state and data as

$$\xi_t = (\mu_t, \sigma_t^2) = \mathcal{E}(x_t) \text{ and } \zeta_t = (\mu_t^{\text{obs}}, (\sigma_t^{\text{obs}})^2) = \mathcal{E}_{\text{obs}}(y_t). \tag{12}$$

We conduct data assimilation in the latent space by assimilating the latent data $\zeta_t$ into the latent state $\xi_t$. Mathematically, the latent state $\xi_t$ approximately follows (up to the VAE reconstruction error) the latent dynamical system

$$\xi_t \approx \mathcal{E}(M(\mathcal{D}(z_{t-1}(\xi_{t-1})), \varepsilon_t)) \tag{13}$$

as a result of the full dynamical system in Equation 1 and the VAE with the reparametrization $z_{t-1}(\xi_{t-1}) = \mu_{t-1} + \sigma_{t-1} \cdot \epsilon_{t-1}$ at time $t-1$. Similarly, the latent data $\zeta_t$ approximately satisfies

$$\zeta_t = \mathcal{E}_{\text{obs}}(H(\mathcal{D}(z_t(\xi_t)) + \gamma_t) \tag{14}$$

as a result of the observation map in Equation 2 and the VAE with the reparametrization $z_t(\xi_t) = \mu_t + \sigma_t \cdot \epsilon_t$ at time $t$. When calculating the likelihood of the observation in the latent space, we simply apply Equation 14 and use an additive latent observation noise $\hat{\gamma}_t$ estimated from the full space observation noise $\gamma_t$ through the observation encoder. Since the coupled encoders aim to match the full states and the sparse observations in their latent representations, we can further approximate the latent observation map in Equation 14 as an identity map, i.e., $\zeta_t \approx \xi_t + \hat{\gamma}_t$, which simplifies and accelerates the computation of the gradient of the log-likelihood function by avoiding automatic differentiation through the map in Equation 14 with respect to the latent state.

Given that the latent state follows the latent dynamics and using the approximate latent observations, we consider the Bayesian filtering problem in the latent space by defining the prediction step as

$$\textbf{Prediction: } P(\xi_t|\zeta_{1:t-1}) = \int P(\xi_t|\xi_{t-1})P(\xi_{t-1}|\zeta_{1:t-1})d\xi_{t-1}, \tag{15}$$

with the transition probability $P(\xi_t|\xi_{t-1})$ determined by the latent dynamical system in Equation 13. The update step for the posterior of the latent state $\xi_t$ given the latent data $\zeta_t$ is defined as

$$\textbf{Update: } P(\xi_t|\zeta_{1:t}) = \frac{P(\zeta_t|\xi_t)P(\xi_t|\zeta_{1:t-1})}{P(\zeta_t|\zeta_{1:t-1})}, \tag{16}$$

where $P(\zeta_t|\xi_t)$ is the likelihood of the latent data given latent state $\xi_t$, computed via the latent observation map in Equation 14. The latent normalization term $P(\zeta_t|\zeta_{1:t-1}) = \int P(\zeta_t|\xi_t)P(\xi_t|\zeta_{1:t-1})d\xi_t$ is also generally intractable.

We employ the diffusion-based ensemble score filter described in Section 2.2 to address the Bayesian filtering problem in the latent space, replacing the state $x_t$ with the latent state $\xi_t$ and the observation $y_t$ with the latent observation $\zeta_t$. Note that both the latent state $\xi_t$ and the latent observation $\zeta_t$ have the same dimension, $2r \ll d$. One step of the Latent-EnSF is presented in Algorithm 1.

---

**Algorithm 1** One Step of Latent EnSF

---

**Input:** Ensemble of the states $\{x_{t-1}\}$ from distribution $P(x_{t-1}|y_{1:t-1})$ and the observation $y_t$. State encoder $\mathcal{E}$, observation encoder $\mathcal{E}_{\text{obs}}$, and decoder $\mathcal{D}$.
**Output:** Ensemble of the states $\{x_t\}$ from the posterior distribution $P(x_t|y_{1:t})$.
    Encode the state $\xi_{t-1} = \mathcal{E}(x_{t-1})$ and the observation $\zeta_t = \mathcal{E}_{\text{obs}}(y_t)$.
    Run the (stochastic) dynamical system model in Equation 13 from $\{\xi_{t-1}\}$ to obtain samples $\{\xi_t\}$.
    **for** $\tau = \tau_k, ..., \tau_0$ **do**
        Estimate the prior score $\nabla_\xi \log P(\xi_{t,\tau}|\zeta_{1:t-1})$ using Equation 18 and 19 in the latent space.
        Estimate the posterior score $\nabla_\xi \log P(\xi_{t,\tau}|\zeta_{1:t})$ using Equation 7 in the latent space.
        Solve the reverse-time SDE in Equation 6 to generate the ensemble of latent states $\{\xi_{t,\tau}\}$.
    **end for**
    Sample $z_t = \mu_t + \sigma_t \cdot \epsilon_t$ from the ensemble of the latent states $\xi_t = (\mu_t, \sigma_t)$ and samples of $\epsilon_t$.
    Decode the ensemble of latent variables $\{z_t\}$ to the ensemble of the full states by $x_t = \mathcal{D}(z_t)$.

---

Because the sparse observations have been mapped to the same dimension as the state in the latent space, we can circumvent the EnSF's weakness associated with the vanishing gradient of the likelihood. In addition, since the encoder is a flexible neural-network representation, the model can constrain the latent observation map $H_{\text{latent}}$ in the latent space to be the identity function $H_{\text{latent}}(\xi_t) = \xi_t$,

which allows for analytical solutions for the score function without needing automatic differentiation. This enables us to fully leverage the computational speed-up of the EnSF. Finally, the VAE's regularization, which encourages the latent vector to follow $N(0, I)$, simplifies the hyperparameter search of the appropriate noise $\gamma_t$. We also make some adjustments to the EnSF to handle numerical instability issues for small-scaled variables, as detailed in Appendix C.

# 4 EXPERIMENTS

In this section, we illustrate the limitations of using the EnSF for sparse observations and demonstrate the improved accuracy and fast convergence of the Latent-EnSF compared to several data assimilation methods for a synthetic complex physical system modeled by shallow water equations with sparse observations in both space and time. We also highlight the merits of Latent-EnSF for medium-range weather forecasting using real-world data and an ML-based dynamical system.

## 4.1 SHALLOW WATER WAVE PROPAGATION

We consider the propagation of shallow water waves described by the shallow-water equations (Vreugdenhil, 1994), a system of hyperbolic PDEs that capture free-surface fluid flow through the conservation of momentum and mass, with three state variables: the water height and the two components of the water velocity field. Such models find practical applications in hydrology for predicting flood waves and in oceanography for modeling tsunami wave propagation.

In the experiment, we consider water wave propagation with the initial water displacement modeled as a local Gaussian bump perturbation from a flat surface in a square domain discretized on a uniform $150 \times 150$ grid, adapted from jostbr (2019). We run the simulation for 2000 steps; specific parameters are provided in Appendix B. Figure 4 displays the evolution of the water height and its pointwise sparse observations with data reduction factors of 4x ($75 \times 75$) and 225x ($10 \times 10$).

For the data assimilation, we start with an initial state for which the Gaussian bump perturbation is shifted away from the true initial state, as shown in the top of Figure 4. We then generate an ensemble of samples (100 by default) by adding a small amount of noise to this initial state. Data assimilation is performed with observations every $k$ (20 by default) simulation steps, unless otherwise stated. We use 100 forward Euler steps to solve the reverse-time SDE, see more details in Appendix A.

### 4.1.1 ENSF AND SPARSITY

In Section 2.3, we illustrated the vanishing gradient of the log-likelihood function for the dimensions in which observations are not available. Here we empirically demonstrate this effect on EnSF for data assimilation with sparse observations at different levels of sparsity.

In Figure 5, we plot the performance in terms of relative root mean square error (RMSE) of the assimilated state compared to the true state as the observation space becomes increasingly sparse. We can observe that increasing the sparsity (from no reduction, to 4x and 225x reduction) of the observation data leads to increasing assimilation errors and slower convergence to the true state by EnSF with 20 samples in the ensemble. When the observation data are reduced by a factor of 225 times, the RMSE of EnSF does not decrease in time, even though we assimilate densely in time (every simulation time step) contrasted to every 20 time steps. In contrast, the Latent-EnSF achieves small relative errors with fast convergence even in the case of extreme sparsity, with the errors at the same level achieved by EnSF with the observation of the full state. These findings align with Bao et al. (2024a) where the sparsity in their experiments for EnSF is limited to a 2x data reduction.

When we increase the number of samples from 20 to 1000 for EnSF with 225x data reduction, we can observe decreasing assimilation errors in time from Figure 5. This is likely due to that the prior score plays a more important role compared to the likelihood function with more sparse observations, and increasing number of samples lead to more accurate estimates of the prior score.

### 4.1.2 LATENT-ENSF RESULTS

We train the coupled VAE as shown in Figure 3 using convolutional layers with input dimension $3 \times 150 \times 150$ and latent dimension $4 \times 10 \times 10 = 400$ with 4 channels. The following experiments

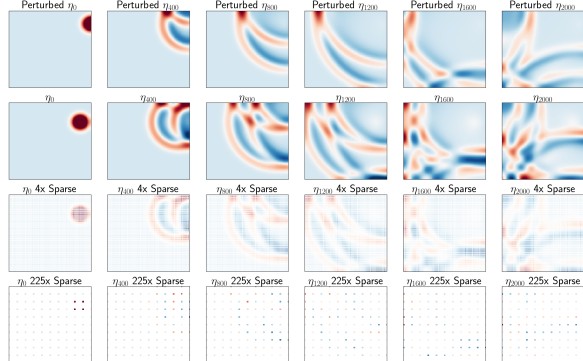

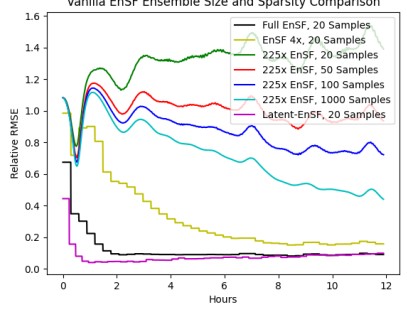

Figure 4: Evolution of states and sparse observations. 225x sparsity markers are enlarged for ease of view.

Figure 5: Relative RMSE of EnSF and Latent-EnSF for observations with different sparsity levels and number of samples.

use a $10 \times 10$ observation grid, corresponding to the 225x sparsity in Figure 5. We compare the Latent-EnSF to the Latent-LETKF and the Latent-EnKF which share the same VAE weights as the Latent-EnSF (Figure 6). For the approximate $\gamma_t$ observation noise terms in the latent space, we adopt a heuristic by taking the mean standard deviation values given by encoded latent states (more details in Appendix C). We additionally compare against a Vanilla LETKF which has desirable properties to handle sparse observations. Our approach is much better in terms of assimilation speed, accuracy, and efficiency (see Table 1) for all cases, see higher observation noise in Appendix D.

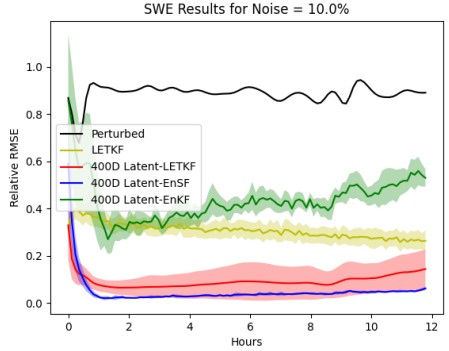

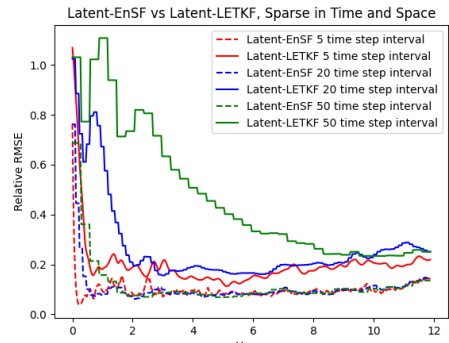

Figure 6: Relative RMSE of Latent-EnSF compared to baselines with 10% observation noise.

Figure 7: Relative RMSE of Latent-EnSF vs Latent-LETKF for varying temporal sparsity.

In Figure 7, we report the comparison of Latent-EnSF and Latent-LETKF for their performances with different temporal sparsity in the observations, which come in every 5, 20, and 50 time steps with increasing sparsity. As we can observe, the assimilation accuracy of Latent-EnSF is not affected by the increased sparsity; only the burn-in time is slightly increased. In contrast, the Latent-LETKF is very sensitive to the temporal sparsity, with much slower convergence for larger sparsity, while Latent-EnSF is robust to sparse observations not only in space but also in time in this experiment.

Figure 8 shows the comparison of the assimilated results of sparse observations using different methods at the first time step, time step 1000, and 2000 (the final time step), and we provide an extended version in Appendix B. We can observe that even though the perturbed dynamics is very far from the ground truth, the Latent-EnSF is able to push the perturbed dynamics to match the ground truth very well with small error, while EnSF fails to do so with its error very close to the difference between the perturbed and true states. Moreover, Latent-EnSF performs the best among the data assimilation methods in the same latent space.

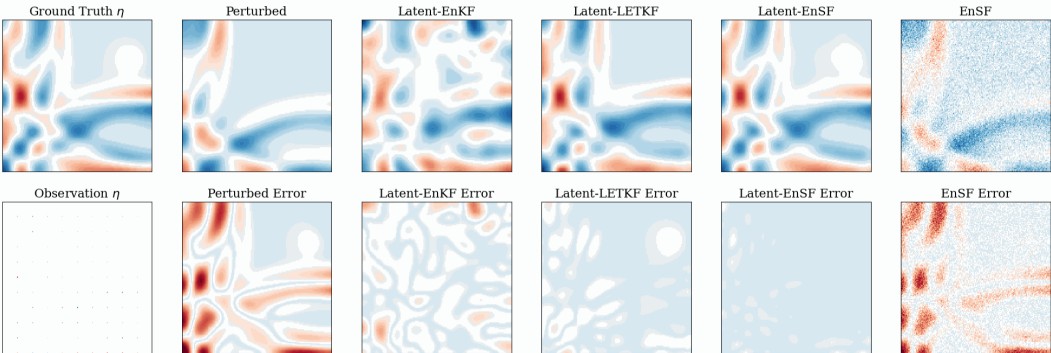

Figure 8: Comparison of different methods at time step 2000, with the ground truth, perturbed states, and assimilated states on the top and their corresponding errors from the ground truth on the bottom.

## 4.2 MEDIUM-RANGE WEATHER FORECASTING

Sparse data assimilation is particularly important for accurate medium-range weather forecasting due to the complexity of the problem which can skew results of machine learning models. The commonly used dataset for conducting these weather forecasting experiments is the ERA5 (Hersbach et al., 2020), coming from the European Centre for Medium-Range Weather Forecasts (EMCWF). This dataset comes with 40 years of reanalyzed data, and can support various degrees of fidelity. Currently, ML-based prediction approaches such as FourCastNet (Pathak et al., 2022), Pangu Weather (Bi et al., 2023), and GraphCast (Lam et al., 2023) have shown to be extremely efficient when applied to weather forecasting compared to standard PDE-based approaches. We adopt the 21 variables used in the FourCastNet paper and train a FourCastNet model on our coarse subset of the dataset for forecasting. When training the VAE for data assimilation, we create a ResNet-style architecture similar to what we constructed in the shallow-water experiments. The encoder downsamples by a factor of 4, resulting in a latent dimension size of $32 \times 36 \times 18 = 20736$. We use this as a demonstrative task to show the feasibility of applying Latent-EnSF to complex systems, so we train and test on 10 years of data with $2.5°$ resolution, with a grid of size $144 \times 72$, as a proof of concept. Evaluations are by averaging the metrics across nine 41-day windows. Data assimilation is conducted once a day, incorporating observations at a 64x sparsity rate, i.e. a grid of $18 \times 9$ observations.

As shown in Figure 9, the forecasting by the FourCastNet-based ML model without data assimilation becomes increasingly inaccurate because of the accumulated errors, which limits its relatively reliable prediction for the first 14 days. The data assimilation errors by EnSF follow closely those of FourCastNet forecasting without data assimilation, which implies that EnSF fails with the data assimilation with sparse observations. In contrast, the Latent-EnSF approach quickly levels off the error in the first couple of days and preserves the accuracy for all the following days, despite the increasing inaccuracy of the FourCastNet forecasting. Note that the Latent-LETKF leads to increasing assimilation errors, much higher than Latent-EnSF, with increasingly inaccurate FourCastNet forecasting. To demonstrate that the FourCastNet is helpful in the data assimilation, we also run the Latent-EnSF without a known dynamical model, in which case we assume the dynamics of the state does not evolve, i.e., $x_{t+1} = M(x_t) = x_t$. From the comparison of Latent-EnSF No Model and Latent-EnSF (with FourCastNet), we observe that the ML-based FourCastNet forecasting does help for the data assimilation, even though it becomes increasingly inaccurate in the long run.

We see in Figure 9 that the assimilation for the state Z500 is much more accurate than that of U500, which can also be observed in Appendix E after 41 days of forecasting by FourCastNet and by data assimilation. This is due to the much higher state complexity with rich local characteristics and sharp changes in the u-component of the wind velocity U500, which would demand more data (with less sparsity) beyond the very sparse $18 \times 9$ observations for more accurate data assimilation.

## 4.3 COMPUTATIONAL EFFICIENCY

To examine the computational efficiency, we report the mean and standard deviation of the time it takes the same machine (Nvidia A6000 + AMD 7543) for one step of the assimilation by each

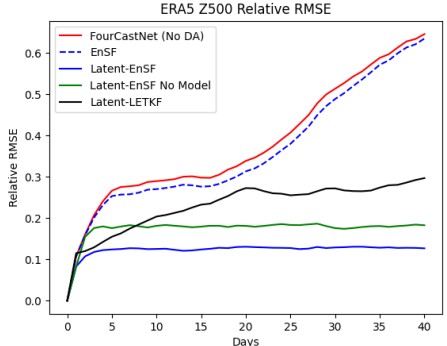 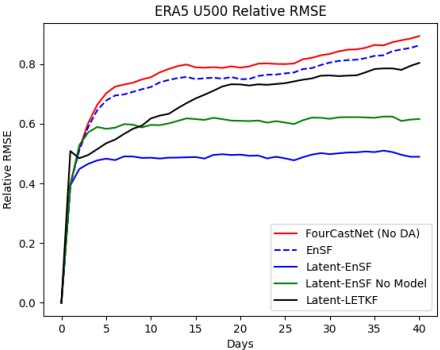

Figure 9: Relative RMSE for the assimilation of Z500 (geopotential at 500hPA) and U500 (u-component of wind at 500hPA) on the ERA5 with respect to the number of days of forecasting.

of the methods in Table 1 for the two applications. The latent dimension $2r$ is applicable to all methods except EnSF with dimension $d$ in the full space. The computational cost of the Latent-LETKF (parallelized with 64 cores) increases approximately linearly with respect to the dimensions compared to the much larger increase of the Latent-EnKF. In comparison, the Latent-EnSF achieves the highest accuracy, fastest convergence, and still uses the smallest time. By comparison of the two applications, we see that although the latent dimensions differ by a scale of 50, the Latent-EnKF takes over 700 times as long for assimilation in the higher latent dimension, compared to 68 times for Latent-LETKF and only 4 times for Latent-EnSF, which implies the much better scalability of the Latent-EnSF compared to the other methods. Remarkably, even with the added complexity of doing a forward pass through the encoder and decoder, the assimilation by Latent-EnSF is still faster than that by EnSF in the full space, making Latent-EnSF suitable for real-time data assimilation.

Table 1: Mean and standard deviation of wall-clock time in seconds of different assimilation approaches for both our experiments, obtained using Nvidia A6000 GPU + AMD 7543 CPU.

| Dataset | Latent Dim | Latent-EnKF | Latent-LETKF | Latent-EnSF | EnSF |
|---------|-----------|-------------|--------------|-------------|------|
| SWE | 400 | $0.056 \pm 0.002$ | $0.103 \pm 0.001$ | $0.051 \pm 0.001$ | $0.223 \pm 0.002$ |
| ERA5 | 20736 | $41.161 \pm 0.640$ | $7.034 \pm 0.055$ | $0.198 \pm 0.008$ | $0.501 \pm 0.001$ |

## 5 CONCLUSION

In this paper, we introduced Latent-EnSF, a novel data assimilation method that addresses the joint challenges of high state dimensionality and high observation sparsity in nonlinear Bayesian filtering problems—key issues for existing approaches. We proposed a coupled VAE for information compression into the latent space, which is trained to consistently match the full states and sparse observations in their latent representations while enforcing latent regularization and ensuring consistent reconstruction of the full states. We also presented the corresponding approximate latent dynamical system and a simplified latent observation map to accelerate the sampling process in the latent space. We demonstrated the high accuracy, fast convergence, and efficiency of the proposed method compared to other approaches for challenging data assimilation problems using both synthetic data and real-world data that are sparse in both space and time.

For further avenues of interest, extending Latent-EnSF to incorporate latent dynamics (Xiao et al., 2024) could further enhance computational efficiency. Additionally, exploring more complex architectures could enable data assimilation for unstructured observation points and continuous-time data where extremely sparse observations occur in a stochastic manner. Overall, Latent-EnSF presents a promising direction for advancing data assimilation techniques and offers an efficient way to utilize machine learning models to simulate real-world systems with improved accuracy.

ACKNOWLEDGEMENT

This work is partially supported by NSF grant # 2325631, # 2245111, and # 2245674. We acknowledge helpful discussions with Prof. Felix Herrmann, Dr. Jinwoo Go, and Pengpeng Xiao.

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

## A    ENSF AND ALGORITHM FOR ONE STEP ASSIMILATION

The EnSF method (Bao et al., 2024b) uses $T = 1$ and the following drift and diffusion terms

$$f(x_{t,\tau}, \tau) = \frac{d \log \alpha_\tau}{d\tau} x_{t,\tau} \text{ and } g^2(\tau) = \frac{d\beta_\tau^2}{d\tau} - 2\frac{d \log \alpha_\tau}{d\tau}\beta_\tau^2, \tag{17}$$

with $\alpha_\tau = 1 - \tau(1 - \epsilon_\alpha)$ and $\beta_\tau^2 = \epsilon_\beta + \tau(1 - \epsilon_\beta)$ for two small positive hyperparameters $\epsilon_\alpha$ and $\epsilon_\beta$, set to $\epsilon_\alpha = 0.05$ and $\epsilon_\beta = 0$ in our experiment, which aid in avoiding the collapse of the generated samples. This choice leads to the distribution $x_{t,\tau} \sim N(\alpha_\tau x_{t,0}, \beta_\tau^2 I)$ conditioned on $x_{t,0} = x_t$, which results in the prior score function

$$\nabla_x \log P(x_{t,\tau}|y_{1:t-1}) = \nabla_x \log \int P(x_{t,\tau}|x_t)P(x_t|y_{1:t-1})dx_t$$
$$= \int -\frac{x_{t,\tau} - \alpha_\tau x_t}{\beta_\tau^2}\omega(x_{t,\tau}, x_t)P(x_t|y_{1:t-1})dx_t, \tag{18}$$

where the gradient $\nabla_x$ is taken with respect to $x_{t,\tau}$, and the weight $\omega(x_{t,\tau}, x_t)$ is given by

$$\omega(x_{t,\tau}, x_t) = \frac{P(x_{t,\tau}|x_t)}{\int P(x_{t,\tau}|x_t')P(x_t'|y_{1:t-1})dx_t'}. \tag{19}$$

Both the score function in Equation 18 and the weight in Equation 19 can be evaluated by sample average approximation with random samples from the distribution $P(x_t|y_{1:t-1})$ obtained in the prediction step in Equation 3.

---
**Algorithm 2** One Step of EnSF

---
**Input:** Ensemble of the states $\{x_{t-1}\}$ from distribution $P(x_{t-1}|y_{1:t-1})$ and the observation $y_t$.
**Output:** Ensemble of the states $\{x_t\}$ from the posterior distribution $P(x_t|y_{1:t})$.
  Run the (stochastic) dynamical system model in Equation 1 from $\{x_{t-1}\}$ to obtain samples $\{x_t\}$.
  **for** $\tau = \tau_k, ..., \tau_0$ **do**
    Estimate the prior score $\nabla_x \log P(x_{t,\tau}|y_{1:t-1})$ using Equation 18 and 19.
    Estimate the posterior score $\nabla_x \log P(x_{t,\tau}|y_{1:t})$ using Equation 7.
    Solve the reverse-time SDE in Equation 6 to generate the ensemble $\{x_{t,\tau}\}$.
  **end for**

---

## B    ADDITIONAL DETAILS ON SHALLOW WATER EQUATIONS

We consider size $L \times L$ with $L = 10^6$ m in each direction and a constant depth $h = 100$ m for the topography of the floor for the shallow water wave propagation. The simulation is run for 2000 time steps using a upwind scheme with a time step $\Delta t = 0.1\frac{\Delta x}{\sqrt{gh}}$ with $\Delta x = L/150$ to satisfy the Courant–Friedrichs–Lewy condition (Courant et al., 1967), see Figure 4 for the evolution of the height and its sparse observations with 4x ($75 \times 75$) and 225x ($10 \times 10$) data reduction.

## C    PINNING DOWN THE LATENT NOISE

When applying the Latent-EnSF, we do not have ground-truth values for the latent observation noise. To obtain an approximate estimate, we calculated the corresponding noise of the latent state by encoding noisy full states. In Figures 11 and 12, we calculate the distribution of the latent error between the true latent state (given by encoding the non-noised state) and the latent encoding of the noisy observations for two different levels of noise. Note that the standard deviation of the latent noise (calculated across ten trajectories) is stable across all time steps, and the mean is approximately zero.

In practice, when applying the EnSF to problems where the observation noise is small, specifying the corresponding observation noise such that $\gamma_t \ll 0.05$ can result in numerical instability issues when sampling the posterior distribution by reverse time SDE with the drift and diffusion terms given in Equation 17.

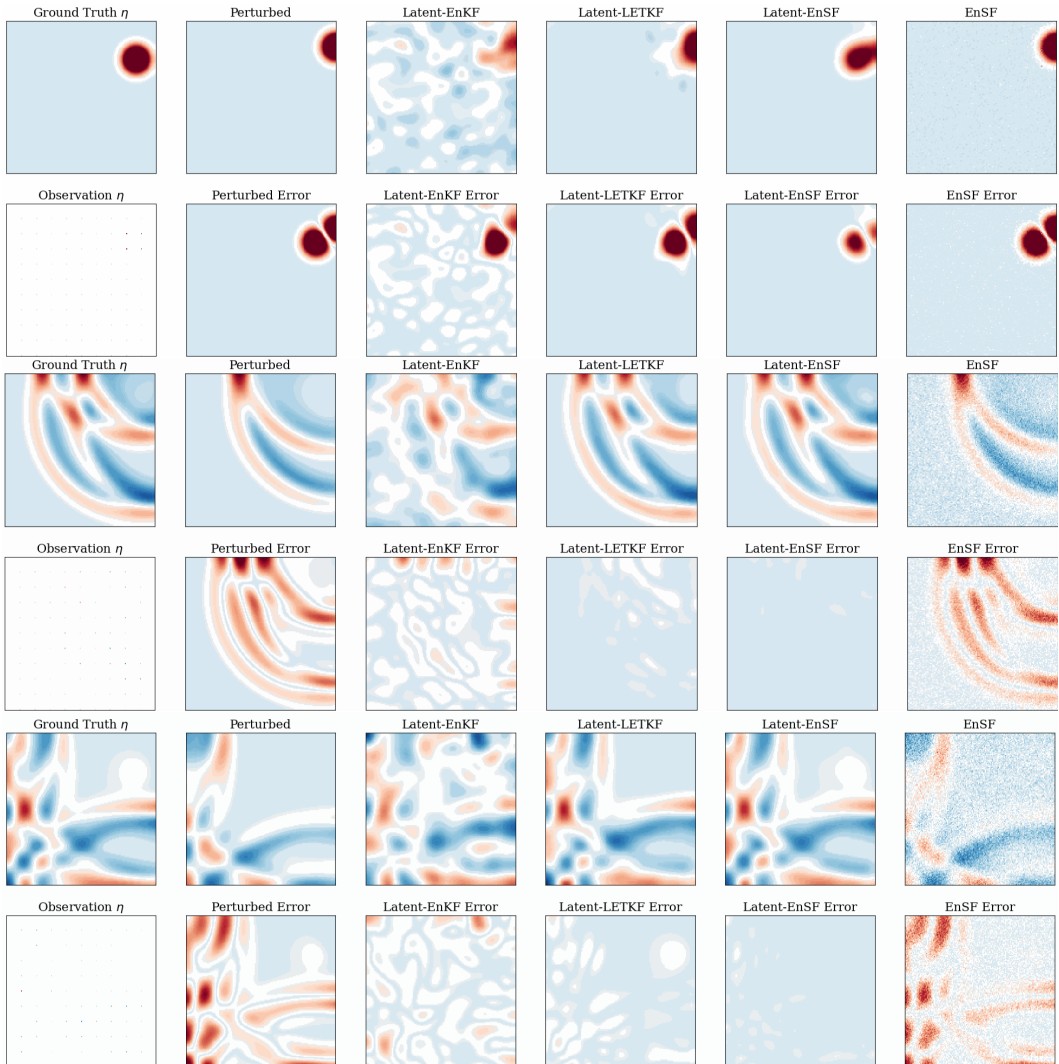

Figure 10: Comparison of different methods at time step 1 (top two rows), 1000 (middle two rows), and 2000 (bottom two rows), with the ground truth, perturbed states, and assimilated states on the top and their corresponding errors from the ground truth at the bottom of each two rows.

This numerical instability issue is also present in the VAE latent space where the representations become compacted as regularization parameter $\lambda$ increases. We address this issue by multiplying the latent representations with a scalar constant $\psi_{\text{latent}}$, conducting the EnSF for the rescaled latent representation, and subsequently scaling the posterior samples back to the original latent representation. We conducted a small grid search experiment when working with the shallow water wave propogation problem in Section 4.1, where $\psi_{\text{latent}} = 20$ seemed to work reasonably.

Remarkably, this $\psi_{\text{latent}}$ applies well in all our experiments, which implies that this scaling hyperparameter stays fairly steady across all problems. Intuitively, this is consistent; the latent regularization conducted by the VAE constrains the latent space to be in a similar range. In contrast, when EnSF is conducted on the full state space, such advantages are lost and a different scaling parameter $\psi$ is needed for different problems with different scales of the state and observation noise.

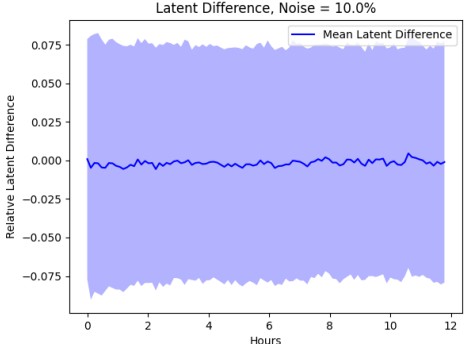
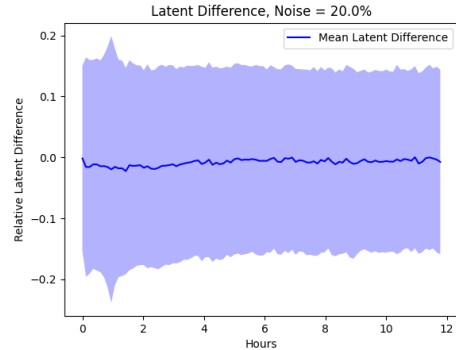

Figure 11: Difference of latent state encoded from full state and latent observation encoded from sparse observation with 10% observation noise.

Figure 12: Difference of latent state encoded from full state and latent observation encoded from sparse observation with 20% observation noise.

# D NOISY RESULTS FOR SHALLOW WATER EQUATIONS

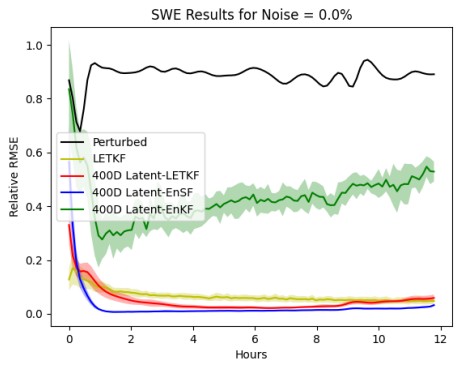
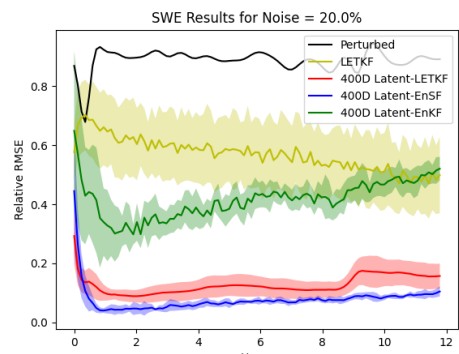

Figure 13: Relative RMSE of Latent-EnSF compared to baselines with no observation noise.

Figure 14: Relative RMSE of Latent-EnSF compared to baselines with 20% observation noise.

# E  SAMPLES FOR WEATHER FORECASTING

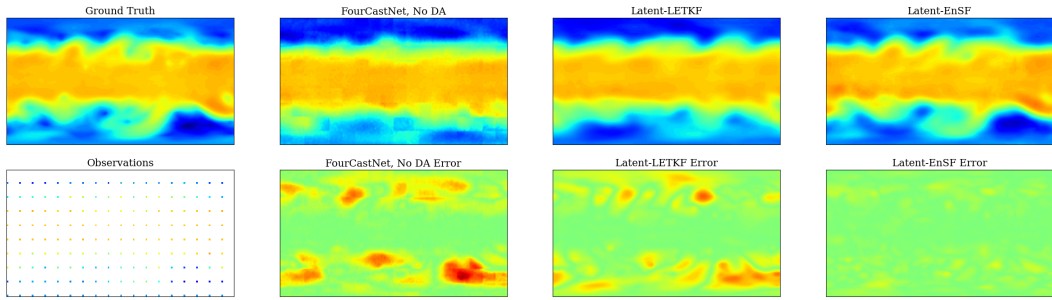

Figure 15: ERA5 Z500 medium-ranged weather forecasting samples after 41 days, along with the errors. Data assimilation is conducted once a day with 64x sparse observations. We compare against a baseline FourCastNet model and Latent-LETKF.

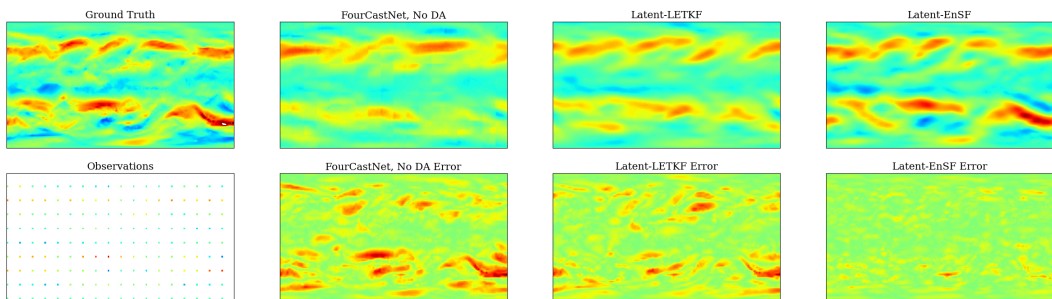

Figure 16: ERA5 U500 medium-ranged weather forecasting samples after 41 days, along with the errors. Data assimilation is conducted once a day with 64x sparse observations. We compare against a baseline FourCastNet model and Latent-LETKF.

