# OpenReview forum: "Latent-EnSF: A Latent Ensemble Score Filter for High-Dimensional Data Assimilation with Sparse Observation Data"
_ICLR.cc/2025/Conference — ICLR 2025 Poster_

### Official Review · Reviewer_KEVJ · 2024-10-31

**Soundness:** 2
**Presentation:** 2
**Contribution:** 2
**Rating:** 6
**Confidence:** 4

**Summary:**

This paper introduces Latent-EnSF, a data assimilation method designed to address the challenges of high-dimensional state spaces and sparse observations in nonlinear Bayesian filtering problems. By extending the Ensemble Score Filter (EnSF) with a latent space representation using coupled variational autoencoders (VAEs), the method transforms both full states and sparse observations into a shared encoded latent space. Then, it leverages EnSF, a score-based filtering method, to filter in the encoded latent space.  The paper claims that leveraging the proposed method can effectively overcome the limitations posed by sparse observations. The numerical results show the improved accuracy, faster convergence, and enhanced computational efficiency in high-dimensional nonlinear systems, with applications in shallow water wave propagation and medium-range weather forecasting.

**Strengths:**

The paper presents a clear and well-motivated rationale for addressing high-dimensional data assimilation with sparse observations in EnSF. The proposed method is intuitively and logically structured, and the experimental results validate its effectiveness and improvements over some of the existing approaches.

**Weaknesses:**

A potential weakness of this paper is the absence of certain important details regarding the developed method, which are noted in the discussion section.

**Questions:**

1. The primary question I have regarding this paper is about the training of the VAE. Specifically, it is unclear whether the encoders and decoder are trained at each time step $t$ or if they are pre-trained. Based on Equations (11–13) and Figure 1, it appears that the input data $ x_t$ for the encoder is derived iteratively during the data assimilation process (from predictive distribution), suggesting that the VAE might need to be retrained at each step. If so, this would significantly slow down the filtering process, raising questions about the efficiency claims presented in the experimental results. Additional clarification on the training procedure of the VAE is necessary to understand how the reported computational efficiency is achieved.

2. The presentation of Section 3.2 is somewhat confusing, and a few points could use further clarification:

   - Could you please elaborate on the need for introducing $ \xi_t $ and $ \zeta_t $? Are these variables different from the $ z_t $ and $ z_t^{\text{obs}} $ used in Section 3.1, and if so, how?
   - In lines 320–321 on page 6 and in Algorithm 1, why is the reparameterization trick needed again to sample the latent variable $ z_t $? My understanding is that after solving the reverse-time SDE in Equation 6, you already have the ensemble of latent states as samples. Could you clarify this step?

3. Minor Issues:

 - Consider moving Figure 1 to Section 3, as it would better illustrate the proposed method in context.

 - In Section 3.1, certain notations are unclear, such as $ \Sigma_t $ and $ \sigma_t $. Clearer definitions would improve readability.

 - On page 5, you note that the GLA proposed by Cheng et al. (2023) can face computational issues when the latent space is high-dimensional. While I agree, typically, as shown in this paper, the latent dimension $ 2r \ll d $, so GLA’s efficiency may not necessarily be compromised in this case. Did you consider including a comparison with GLA in the experiments?

 - While the method is based on the EnSF, it seems that the discussion on the existing data assimilation methods for high-dimensional state spaces and sparse observations is missing e.g. (and more),

      - Chen Y, Sanz-Alonso D, Willett R. "Reduced-order Autodifferentiable Ensemble Kalman Filters," *Inverse Problems*, 2023; 39(12):124001.

     I believe including the discussion of more related work could make the proposed method's positioning more comprehensive.

---

> ### Author Response · Authors · 2024-11-18
>
> We thank the reviewer for their detailed feedback and questions! Below we have provided some additional details and explanations for the reviewer's points in addition to the text in the main rebuttal:
> # Training of the VAE
> We refer to "Training process of Latent-EnSF" of the main rebuttal.
>
>  # Introducing Variables
> $z$ refers to the samples from the latent representation (after the reparameterization trick of the VAE), whereas $\xi$ and $\zeta$ are the mean variance combinations of the state and observations respectively.
>
> # Reparameterization Trick and Resampling
> Empirically, we found that using the mean by itself seems to achieve better results, which we will reflect in changes to the manuscript. As a result, the reparameterization trick is now only used in training time to ensure sufficient smoothness of the latent representation.
>
> # Minor Issues
> Thanks so much for pointing these out, will fix and clarify in the manuscript.
>
> # Comparison against GLA
> We note that GLA’s results are approximately encapsulated within the Latent-EnKF results we show when the observation function is linear (as tested for the shallow-water equations). As a result, on high-dimensional latent spaces, the issues plaguing Latent-EnKF should still occur as in the dimension experiment in Figure 10.
>
> # Discussion on Existing Data Assimilation Approaches for High-Dimensional Systems
> Thanks for the paper suggestions! We will add more discussion on this topic in the revised manuscript.

---

> > ### Author Response · Authors · 2024-11-24
> >
> > Dear Reviewer KEVJ,
> >
> > Thank you for your time and effort in reviewing our paper and helping us polish it with your insightful feedback. As the end of discussion period draws near, we would be grateful you could let us know you have received our responses, and if they have addressed your concerns and improved your evaluation of the paper. Please don't hesitate ask if you have any further questions.
> >
> > Thank you very much!
> >
> > Sincerely,
> > Authors

---

> > > ### Comment · Area_Chair_F8xV · 2024-11-26
> > >
> > > Dear reviewer,
> > >
> > > Please make to sure to read, at least acknowledge, and possibly further discuss the authors' responses to your comments. Update or maintain your score as you see fit.
> > >
> > > The AC.

---

> > ### Comment · Reviewer_KEVJ · 2024-11-26
> >
> > Thank you for the clarifications. However, I still have some questions regarding the training of the VAEs. My understanding is that the VAE is trained using the objective function outlined in Eqs. 11--13. If I understand correctly, the training data consists of underlying (state, observation) pairs, with the states being sampled from the true state posterior distributions.
> >
> > However, during inference, as seen in Fig. 1, the inputs (latent states) for the state encoder appear to be derived from the predictive distribution $x_t \vert y_{1:t-1}$. Does this discrepancy have any significant impact on the model's performance?
> >
> > Additionally, what happens if the latent matching is exact, meaning the norms in Eq. (13) are zero? Is this scenario possible, and if so, how does it affect the latent EnSF?

---

> > > ### Author Response · Authors · 2024-11-26
> > >
> > > We thank the reviewer for responding to our rebuttal.
> > >
> > > > However, during inference, as seen in Fig. 1, the inputs (latent states) for the state encoder appear to be derived from the predictive distribution $x_t | y_{1:t-1}$. Does this discrepancy have any significant impact on the model's performance?
> > >
> > > The reviewer strikes upon the key objective of conducting data assimilation by the Bayesian Filtering method, which is to update samples from the predictive distribution $P(x_t | y_{1:t-1}$ to the posterior distribution $P(x_t | y_{1:t})$ by means of incorporating the likelihood $P(y_t | x_t)$. The machine learning model is fairly robust to small perturbations due to the regularization (even if minimal) of the latent space, which allows for slightly perturbed observations (as in the case with the noisy ablation experiments in Figures 1R-5R) and states (when encoding the prior) to be corrected.
> > >
> > > > Additionally, what happens if the latent matching is exact, meaning the norms in Eq. (13) are zero? Is this scenario possible, and if so, how does it affect the latent EnSF?
> > >
> > > If the latent matching is always exact (in every possible scenario), that means that there is an exact mapping from the observations to the states for any trajectory, which is an unrealistic assumption. There are certain instances during data assimilation where the latent representation would be very close to the true state representation, but that is due to the latent observations being a consistent distribution centered around the true latent state (Figure 6-7R).

---

> > > > ### Comment · Reviewer_KEVJ · 2024-11-27
> > > >
> > > > I would say that the states from the prior (prediction) distributions are not merely small perturbations of the states from the posterior (filtering) distributions. The posterior distribution can significantly diverge from the prior distribution, depending on the new evidence introduced.

---

> > > > > ### Author Response · Authors · 2024-11-27
> > > > >
> > > > > The reviewer brings up a good point in that it's sometimes not just small perturbations in the prior; our experiments with water equations actually starts off with a very mis-estimated prior and is still able to reduce the error by the means of data assimilation.
> > > > >
> > > > > Also, we realize that we misread the previous response when writing our folllow-up. Though the assimilation process tries to update to the posterior $p(x_t | y_{1:t})$, when actually training the VAE, there is no "posterior" distribution (in context of conditioning on the observations). It simply encodes the distribution of states possible states $x_t$ from $t \in [0,T]$ where $T$ is the max time step.

---

> ### Comment · Reviewer_KEVJ · 2024-11-27
>
> Let me elaborate for a more effective discussion. Suppose you have a set of data (state, observation) pairs that you've collected from a certain system. Let's say this system is a linear Gaussian state-space model (LGSSM), WLOG.
>
> The states and observations correspond one-to-one. So, can I equivalently understand that the state trajectories you sampled come from a true underlying posterior distribution?
>
> You trained a VAE based on states sampled from this posterior distribution, but during inference, the states inputted into the VAE come from the prior (prediction) distribution, which is quite unnatural to me.
>
> The authors claim that the empirical results are good, but why is that?

---

> ### Author Response · Authors · 2024-11-27
>
> We thank the reviewer for their follow up.
>
> > the states and observations correspond one-to-one. So, can I equivalently understand that the state trajectories you sampled come from a true underlying posterior distribution?
>
> The states and observations do not correspond one-to-one, but instead, many-to-one, especially in particularly chaotic systems. The state trajectories used during training are sampled from the evolution dynamics $P(x_t | x_{t-1})$ as outlined in Equation 1 in the paper without explicit conditioning on the observations $y_t$. When we progress the model dynamics, we equivalently model the corresponding $P(x_t | x_{t-1})$ (either through machine learning models, or through numerical simulation of the physical system).
>
> Additionally, note that the main formulation to encode the corresponding prior states comes from the general branch of Latent Assimilation models, which use VAEs in context of Bayesian Filtering problems.
>
> > The authors claim that the empirical results are good, but why is that?
>
> The VAE is trained to encode a wide range of possible states and observations into a shared latent space through data trained on multiple trajectories, ensuring that the representation remains valid even under predictive dynamics. Additionally, take note of the results in Figure 4R-5R. Here, the lower bound on the error is for the Full Reconstruction, which is the error given by the reconstruction of the true un-noised state. The rest of the error for the Latent-EnSF is compounded by the uncertainty given by the noise, the lack of information due to the sparsity of the observations, and errors given by the prior term. However, note that the observation reconstruction results are worse than the assimilated results. This shows that even if there is some error introduced in the prior, the added information still plays an important role in the data assimilation process.

---

> > ### Comment · Reviewer_KEVJ · 2024-11-28
> >
> > Thank you for the response. I was quite misled by Figure 1, which outlines the inference process, but I now understand the training aspects, particularly how data pairs are extensively sampled from the system.
> >
> > Despite this clarification, I still feel that the training of the VAE lacks detail, a concern also reflected in other reviewers' comments. Additionally, the motivation behind the design of the VAE-inspired loss function could be more clearly explained.
> >
> > I have raised my score as my primary concerns have been addressed. However, regarding presentation flow and overall quality, I still find the paper just below the threshold for acceptance.

---

> > > ### Author Response · Authors · 2024-11-28
> > >
> > > We thank the reviewer for their continued discussion on the paper, and are extremely grateful for their contributions in polishing the paper. We will continue to improve upon the paper based on the reviewers' feedback, including more detailed discussions and clarifications and other points throughout the discussion phase.
> > >
> > > Hope you have a great Thanksgiving!

---

### Official Review · Reviewer_se1f · 2024-11-01

**Soundness:** 3
**Presentation:** 3
**Contribution:** 3
**Rating:** 6
**Confidence:** 3

**Summary:**

This paper develops the recently proposed Ensemble Score Filters (EnSF) by incorporating Variational Autoencoder (VAE) to define EnKS in the latent space derived by VAE. The resulted algorithm, hence named latent-EnSF, inherits the effectiveness of EnSF in high-dimensional data assimilation, but also solves the issued of EnSF when dealing with the sparse observations. Despite some possible typos, the proposed method is clearly explained, supported with thorough comparison with various alternative algorithms including vanilla EnSF, latent-EnKF, latent LETKF, FourCastNet using two models in water wave propagation and medium-range weather forecasting.

**Strengths:**

Most of the implementation in the latent space of EnSF is straightforward. The added benefit of handling sparse observations by running EnSF in the latent space is impressive and has potential impact in data assimilation in high dimensions with sparse observations. I certainly appreciate the thorough comparison offered in this paper.

**Weaknesses:**

1. EnSF has already achieved great results in high-dimensional data assimilation problems. VAE is used as a dimension reduction method. The mysterious capability of handling sparse observations by combining them remains largely un-explained. Though there is some on line 323, the paper may benefit in clarity from more elaboration in this aspect.

2. The most difficult part to understand is the latent matching term (13) on line 244. Two encoders, the state encoder and the data encoder, compress two different things. Why do they need to match the codes (mean and variance) in the latent space? I see the authors report segregated latent representations without such term, but I still find it unnatural and lack some explanation on its contribution to the capability of handling sparse observations.

3. The precision of solutions to inverse problems naturally depends on the data information available. While I am impressed by the comparison between EnSF and latent-EnSF in terms of handling the sparse observations, some comments on the limitation of the proposed latent-EnSF is also appreciated.

**Questions:**

* Line 222: What do you mean by "the output as a full-dimensional observation in the latent space"? Is it "in the full space"?

* Equation (14), KL divergence is defined between two distributions. What are they in your equation (14)? Did you mean to say $KL(N(\mu_t, \Sigma_t)|| N(\mu_t^\text{obs}, \Sigma_t^\text{obs}) )? Then pay attention to the signs.

* Line 314: can you comment on the computational cost of this step because equation (16) involves both encoder and decoder.

---

> ### Author Response · Authors · 2024-11-18
> **Rebuttal Reviewer 5**
>
> We thank the reviewer for their detailed feedback and questions! Below we have provided some additional details and explanations for the reviewer's points in addition to the main rebuttal text up above:
>
> # Elaboration on Latent-EnSF Handling Sparse Observations
> The Latent-EnSF is able to make use of information from the training data to inform the distribution of the state given the observations, but the standard EnSF is unable to do that with sparse observations. As a result of learning from the training data, the reconstruction from the VAE could also be regularizing for some of the noise terms which are otherwise present during standard EnSF processes. Our method addresses specifically that in the latent space, there is no vanishing gradient problem due to the encoded observations and ensemble states being in the same space. All of the factors combine to make Latent-EnSF particularly effective for this problem. We will update the manuscript to elaborate on this more.
>
> # Limitations
> We can definitely talk about some of the limitations of the model. Firstly, the future states still needs to be in the generalization area of the VAE; if conditions change drastically, then the data will need to be updated to reflect that change. Because the VAE is able to take advantage of the training data, if its outside of its supported area then the reconstructions can become highly uninformative, as compared to standard EnSF which is data-free.
>
> Secondly, because it requires data to train, this can be much harder for new fields without as much available data where simulations can be quite complex. Generating the trajectories and collecting the data would require a significant amount of start-up time before the model can be integrated.
>
> Finally, if the observation data becomes too sparse (and as a result not informative enough), then with the current architecture our approach cannot capture accurate data assimilation. However, this is not only limited to this model, but other data assimilation filtering approaches as well. We can solve this by adopting alternative architectures, see "Grid Observations and Extreme Cases" for a more detailed analysis.
>
> # What it Means to be a Full-dimensional Observation in the Latent Space
> This point refers to when the state space is some dimension $R^d$, whereas the latent space is in some dimension $R^{2r}$.
> The observations are in some state $R^k$ which is some sparse function of state. As a result, standard data assimilation is done by using observations in $R^k$ to states in $R^d$, with $k<<d$. The state encoder casts the ensemble of states from $R^d$ to $R^{2r}$, and the observation encoder casts the ensemble of states from $R^k$ to $R^{2r}$, which results in the dimensions being the same in the latent space.
>
> # KLD Term
> Thanks for bringing this up, we refer the reviewer to the corresponding section in the main rebuttal post.
>
> # Computational Cost of Line 314
> The main computational cost of this step is in propagating the original model, but that is shared across all latent-assimilation style models. In Table 1, we show the wall clock times which build in the encoding and decoding into the assimilation portion, since the original model propagation is constant across all models.

---

> > ### Author Response · Authors · 2024-11-24
> >
> > Dear Reviewer se1f,
> >
> > Thank you for your time and effort in reviewing our paper and helping us polish it with your insightful feedback. As the end of discussion period draws near, we would be grateful you could let us know you have received our responses, and if they have addressed your concerns and improved your evaluation of the paper. Please don't hesitate ask if you have any further questions.
> >
> > Thank you very much!
> >
> > Sincerely,
> > Authors

---

> > > ### Comment · Area_Chair_F8xV · 2024-11-26
> > >
> > > Dear reviewer,
> > >
> > > Please make to sure to read, at least acknowledge, and possibly further discuss the authors' responses to your comments. Update or maintain your score as you see fit.
> > >
> > > The AC.

---

> > > > ### Comment · Reviewer_se1f · 2024-11-27
> > > >
> > > > Thank the authors for their responses. Many of my questions remain unanswered, e.g. 'The most difficult part to understand is the latent matching term (13) on line 244. Two encoders, the state encoder and the data encoder, compress two different things. Why do they need to match the codes (mean and variance) in the latent space?'. And there is no update in the rebuttal revision or did I miss it? Not completely satisfied with the reply, I will at most keep my score.

---

> ### Author Response · Authors · 2024-11-27
>
> We thank the reviewer for their response and follow up questions!
>
> > The most difficult part to understand is the latent matching term (13) on line 244. Two encoders, the state encoder and the data encoder, compress two different things. Why do they need to match the codes (mean and variance) in the latent space?
>
> The primary purpose of matching the means (we remove the variance term in our new formulation) in the encoders is to provide consistency. The state encoder converts full states to latent representations while the observation encoder maps the sparse observations to latent representations. While these are different sources of information, in essence, they both describe the same physical state. The key of enabling effective data assimilation is that there is a mapping $y_t = H(x_t) + \gamma_t$. Here, we align the means such that $H_r=I$, the identity transformation, ehere $H_r$ id the observation operator in the latent space. Without such regularization, then it leads to mismatches of the latent states and observations, resulting in a failure of the Bayesian filtering process.
>
> > And there is no update in the rebuttal revision or did I miss it?
>
> Due to the large changes needed for the paper revision (for example, the explanation below on the failure of the vanilla EnSF), we are first fully gathering all the material needed for the revised manuscript in the rebuttals for each reviewer before updating the manuscript so that we can focus on the content first rather than the page limitations. The planned changes to the manuscript can be mostly encapsulated in the main rebuttal and the following discussions.
>
> > Some additional info on the "mysterious capability of handling sparse observations" question
>
> Below, we have some updated explanations for the reasoning of the failure of the Vanilla EnSF as a result of our discussion with Reviewer GvWu, which should supplement the clarification on why sparse assimilation works well for Latent-EnSF as compared to vanilla EnSF:
>
> ## Gradients of the Score Clarification
>
> When the observations are sparse, due to the missing gradient entries in the likelihood term, the total $L2$ norm of the gradient of the likelihood for the vanilla-EnSF is now scaled down by a factor of $\sqrt{d/k}$ (state is $d$-dimensional, observations are $k$-dimensional). As a result, the prior term dominates by the same factor as compared to full observations. Although this doesn't detract theoretically from the score filter approach given infinitesimally small time steps, in order for the EnSF formulation to be accurate in a sparse context, it then requires that the estimated prior term $P(x_{t} | y_{1:t-1})$ is somewhat representative of the true $P(x_t)$. There are now two sources of error here:
>
> Firstly, for problems where the initial state is misspecified, the assimilation portion of the original EnSF relies heavily on the likelihood component of the score in the beginning, which can lead to very slow assimilation progress due to the total likelihood gradient norm. With observation data that is also somewhat sparse in time, the assimilation progress may not be fast enough to correct for the error which is propagated across time steps.
>
> Secondly, the prior score term is primarily estimated by Monte Carlo samples. When we have full observations, the Monte Carlo error of the prior is negligible compared to the correction by the likelihood term. However, when the observations are sparse, then due to the shrinking magnitude of the likelihood term gradient, estimation errors within the prior term are magnified. As a result, to achieve good assimilation results, the number of ensemble members also increases greatly, which defeats the purpose of the EnSF being fast and scalable for data assimilation. The Latent-EnSF addresses the information loss by supplementing it with learned generalizations from the training data, which by extension draws the numerical component back to equilibrium as in the vanilla EnSF with dense observations.
>
> ## Experimentation for Above Claims
> In Figure 8R, we show the relative RMSE for the standard EnSF with different ensemble sizes. We assimilate the observation data at each time step (dense in time) as compared to the other approaches which are more sparse in time (every 20 time steps). Note that for small ensemble sizes (5 and 20), the estimation error actually overrides the assimilation progress, causing the error to blow up (error source 2). For high ensemble sizes (500 and 1000) the model does fairly well, but assimilation progress per time step is still remarkably slow (error source 1), even though we assimilate 20 times more than we did for the VAE-based baselines. Therefore, the EnSF needs multiple factors to work with sparse observations: observations dense in time, large ensemble size, and an accurate model, which may not exist in real-world applications.

---

> > ### Comment · Reviewer_se1f · 2024-12-02
> >
> > Thank the authors for their elaboration. Regarding the latent matching terms, I appreciate the explanation that "While these are different sources of information, in essence, they both describe the same physical state." However the following sentence is misleading: $x_t\in \mathbb{R}^d$ and $y_t\in\mathbb{R}^m$ do not even have the same dimensionality (usually $m\ll d$ for sparse observations as you commented), how could you achieve $H=I$ as the identity transformation? It may be helpful in the revision to illustrate in the numerics the effect of latent matching terms (i.e. compare the results with vs without such terms).

---

> ### Author Response · Authors · 2024-12-02
>
> Thanks for the additional questions; we thank the reviewer for pointing out the $H=I$ typo in our previous response. We are referring to the observation operator (previously unnamed) in the latent/reduced space (given by equation 17 in the paper). We have gone back and fixed out previous response for the benefit of future readers.
>
> We can include a section in the appendix for a latent matching term comparison for the future revisions.

---

### Official Review · Reviewer_oZ4F · 2024-11-01

**Soundness:** 4
**Presentation:** 3
**Contribution:** 3
**Rating:** 8
**Confidence:** 4

**Summary:**

The paper aims to address the problem of assimilating sparse observation data effectively using the ensemble score filter framework for data assimilation. Their method proceeds by encoding both state and observations to a latent space of the same dimension before applying the score-based ensemble filter. This allows one to avoid the issue of having mismatched size of the observation and state dimensions, preventing vanishing gradient of the likelihood. Experimental results demonstrate the efficacy of the approach showing further promise of score-based methods for data assimilation.

**Strengths:**

The paper addresses a critical issue with many of the score-based filters being proposed recently, namely, that the observations being assimilated are too idealistic. The methodology proposed in this work brings it closer to more realistic scenarios, where observations are sparsely distributed. Results in the experiments are encouraging, showing that the latent EnSF can significantly outperform standard EnSF or LETKF in latent space. The writing is generally clear and sufficient details are provided to implement the proposed model.

**Weaknesses:**

One key weakness of this work is that baseline comparisons against traditional DA methods are missing. For example, how does it compare to standard ensemble Kalman filter (LETKF) or 4DVar in physical space? The authors consider LETKF in latent space but this approach is not actually used in practice. If we want to really demonstrate efficacy of the approach, a comparison with these traditional method should be provided. These methods are likely to perform quite well on the presented examples so it will help us see how much extra benefit we can gain by replacing them with the proposed latent EnSF framework. Furthermore, some details of the algorithm were unclear to me, which you can find below in __Questions__. Finally, plots to show the sparse observations can be improved by increasing the marker size. At the moment, it is impossible to see where the observations are without zooming in significantly.

**Questions:**

- The authors should clarify that the Kullback-Leibler divergence in (14) is taken with respect to the standard normal measure. When talking about statistical divergences, one should specify which two measures the divergence are taken with respect to and in which order.
- Is the loss in (11)-(13) correct? The reconstruction term should be marginalised over the latent variable $z_t$ if this were to be trained like a VAE.
- Related to the above question, can the loss (11)-(13) be derived from first principles via variational Bayes? Or is the loss more just "VAE-inspired"?
- Do we train for the encoder/decoder at every timestep $t$? Do the authors take into account its training cost if so? It seems quite expensive to achieve in practice.
- At the moment, it seems like the method only works in the case where the observations are placed on a regular grid. Is there a way to extend this to a more realistic scenario where the observations are less regularly sampled?
- Could the authors please elaborate more on how they compute the latent observation noise $\hat{\gamma}_t$? The authors only give a passing remark but this does not seem so obvious to me.
- How do the results in the experiments compare to more traditional data assimilation methods like the EnKF or 4DVar? Does the latent EnSF still outperform these in the sparse observation setting?

---

> ### Author Response · Authors · 2024-11-18
> **Rebuttal Reviewer 4**
>
> We thank the reviewer for their detailed feedback and questions! Below we have provided some additional details and explanations of the reviewer's points, in addition to the main rebuttal text above:
>
> # 4D-Var
> We present our current problem as a filtering problem, which has less data than a smoothing problem such as 4D-Var. As a result, we only compare against other models which have the same amount of information flowing in at each time step.
>
> # Comparison against LETKF
> The reviewer makes an excellent point for comparing a standard LETKF instead of Latent-LETKF for high dimensional linear spaces. Therefore, we have conducted additional experiments on the shallow water equations with the standard LETKF and sparse observations in the main rebuttal response. As in line with other reviewers' requests, we have also included results with differing amounts of noise as well as error bars plotted over 10 different initializations of the problem. From our results, it seems that the latent encoding inherently helps speed up the process of the assimilation, which gets tougher for the standard LETKF as the amount of noise increases.
>
> # KLD Term and VAE Loss
> We thank the reviewer for noting this, and address the KLD Term in the main rebuttal section. We currently express the implementation form of the VAE, which assumes that with enough samples and epochs that it achieves the expectation (marginalization) the latent variable. We will make the revisions in our manuscript. As for the actual modified loss, it is in fact just VAE-inspired, as compared to coming from Variational Bayes. In fact, in our recent experiments, we made some additional changes with adopting a KLD weighting similar to that of Latent Diffusion Models, which we note in the main rebuttal section.
>
> # Encoder and Decoder Training
> When we train the VAE, we train it on offline data. As a result, during assimilation time, there is no need to train the VAE; only forward passes are needed for inference.
>
> # Grid Observations
> We refer the reviewer to the "Grid Observations and Extreme Cases" section in the main rebuttal text.
>
> # Latent Observation Noise Computation
> We refer the reviewer to the "Latent Observation Noise Computation" section in the main rebuttal text.

---

> > ### Comment · Reviewer_oZ4F · 2024-11-22
> > **Thank you for the response**
> >
> > I thank the reviewer for clarifying my concerns, mainly about addressing the sparsity patterns, computation time of the VAE and comparison against a more traditional baseline like the LETKF. I will consider raising my score.

---

> > > ### Author Response · Authors · 2024-11-24
> > >
> > > Dear Reviewer oZ4F,
> > >
> > > Thank you very much for your time and effort in reviewing our paper and helping us polish it with your insightful feedback!
> > >
> > > Sincerely,
> > > Authors

---

### Official Review · Reviewer_GvWu · 2024-11-02

**Soundness:** 3
**Presentation:** 2
**Contribution:** 3
**Rating:** 6
**Confidence:** 4

**Summary:**

The paper identified a shortcoming in the recently developed Ensemble Score Filtering method, which is that observations at sparse sites lead to a collapse in the model gradient, by introducing a low-dimensional latent variable encoding, which recovers improved inference in this sparse setting.

**Strengths:**

This problem is high value, and the approach unifies several interesting ideas.

Empirical results are impressive.

**Weaknesses:**

Minor cosmetic and presentational issues. These are not crucial, but I will attempt to note when I see them

More crucially, theoretical explanation for the empirical success does not seem complete, and details are missing which might allow the reader to deduce this for themselves.

Firstly, there are a number of small errors and typos, e.g. equation 14 introduces the KL "Divergence" as  function of the parameters of the Gaussian distribution $\operatorname{KLD}(\mu,\Sigma)$ .That is not a divergence. It looks like it is maybe the conventional KL between a given diagonal Gaussian and a standard diagonal Gaussian? There are also some in the section 4.1.1, which I will address in the Questions. There are a couple more I noticed but did not note down; I'll leave these cosmetic issues aside for the moment so we can get to the substantive weakness and return if there is time.

The paper's argument on an important assertion, that sparse observations are problematic EnSF updates (l202). However, this part of the argument is not made sufficiently clearly for such a central part of the paper. The first supporting evidence  (figure 8 in appendix b) is confusing - AFAICT it simply shows what a sparse sampling grid looks like, which... doesn't really show anything. That is just a graphic depiction of the form of the observation operator $H$, I think. Section 4.1.1 does heavier lifting, which is that attempts to show that empirically, as the updates get sparser in the sense of sampling a coarse spatial grid, the quality of the EnSF estimates degrades. However, I am not convinced it does show that; It is unclear to me if this is because the experimental evidence itself is lacking; it could be the labeling in the figures. See questions for clarifications on that point.

The explanation for what problem is being solved here and why the solution works seems incorrect or at least incomplete.
If the baseline is EnSF, which uses (up to some discretization error and ensemble approximation etc) the "true" model $M$ and the "true" observation operator $H$, then it should in principle recover the "best" estimate of the system dynamics. the proposed method Latent-EnSF claims to get superior performance by introducing a further layer of approximation, mapping the observations and dynamics into a VAE. Since we have introduces a further approximation then we must assume that all else being equal, the VAE-based method should be worse in accuracy (it might be faster at inference time or something like that).

As such, if Latent-EnSF  it is doing better in accuracy terms than EnSF, it seems that it should be because it gets to exploit more information at inference time than does EnSF. Presumably this "extra" information is in the form of better spatial smoothing encoded in the encoder/decoder networks of the VAE, which can presumably learn the which spatial fields are in the training data. If so, we expect that the information enters the method via the training losses (11,12,13) and also by the training data set. The paper does not seem to interrogate the training losses and also I cannot easily work out what the actual training procedure is for the neural nets, so it is not clear to the reader what smoothing the neural nets are actually learning.

**Questions:**

## What is the actual deficiency of EnSF?

The empirical results are great.
What is actually happening here theoretically? Since Latent-EnSF and EnSF are both targeting the same likelihood, and the VAE-based Latent-EnSF is making a looser approximation to the posterior than the original model, then it is not sufficient IMO to claim it is simply addressing a problem of missing gradient; Latent-EnSF needs to be jsutified in terms of how it draws better inference from the same information.
I've expanded on this question in the _weaknesses_ section, but I invite you to answer it here.

As part of that I also suggest it will be good to clarify which data (training data, inference time observation) informs the final prediction. Questions about that in the next section.

## What is the training distribution?

What is the training process used for the NN? What distribution of states is in that dataset? Does an encoder trained for (however many trajectories are in this dataset) generalize badly to data assimilation problems drawn from a different dataset (say, different initial conditions) in a way that EnSF does not? That would be evidence that this method learns a good spatial smoothing, and would also suggest a deficiency in the original EnSF, if it has difficulty in spatial smoothing

## Section 4.1.1
I'm confused about both figures and the explanation. Can you answer for me the following?

1. Figure 2: I'm not sure what $\eta$ is. Is it the reconstructed state? If so, in the 4x sparse row we see that the reconstructed solution has a banded structure. Why? I know that the _observations_ updates are sparse, but when we propagate from those back through our diffusion steps, should we not recover a densely sampled field at whatever resolution our simulator $M$ runs?
2. Figure 3:

   1. it's not clear what "perturbed" means here. From cross checking I think this is the error that a simulator would accrue from failing  to update at all. Why is it constant? For interesting models (e.g. with lyapunov exponent ) I would expect this to grow over time. This leads me to wonder if the systems that are being modeled here are not very interesting, or maybe that RMSE is a bad measure, or that the duaration over which we observe the system is simply not interesting. For the kind of challenging systems mentioned in the paper (e.g. the high dimensional lorenz) I would expect diverging loss metrics under perturbation
   2. I'm not even sure which model is being plotted in these figures. It looks like a fluid PDE?
  3. There are no error bars here. I would expect for such a central plot that we would make some attempt to plot error bars under random perturbations and ideally also random initial conditions.

## What sparsity patterns are allowed?

The proposed method appears to have a curious limitation, which is that the spatially-sparse observations must occur at fixed sites $S$ for all timesteps. This is not required in the EnKF and, although i know it less well, should not be required in the EnSF. Is it required for this model also, or can that be relaxed?

## section 4.1

>However, though the Latent-LETKF beats the Latent-
>EnKF benchmark, especially with higher latent dimensions, our approach is still much better in
>terms of assimilation speed, accuracy, and efficiency (see Table 1) for all cases.

Some confusing phrasing. *Is* that table 1 shows? It seems to me that it shows on two benchmarks, Latent-EnSF is the fastest in terms of wall-clock inference time. (Side point: it's ok in my opinion to use minimal wall clock time for operations with a deterministic number of FLOPs, means and standard deviations are not ideal for timing execution which is an asymmetric right-skewed distribution who randomness is mostly due to external factors) However the graph shows that on (I think on some different model?) that Latent-EnSF is more accurate than Latent-LETKF.
Moreover, some of these methods (the VAE-based ones) require us to train a neural network, and others (EnKF) do not. How much does the NN training time add to these? When I've trained my encoder networks, does that generalise to unseen initial conditions, or do I need to train a new encoder? When do I need to train a new encoder? How costly is that?
These is not to say that I think anything is incorrect, but rather that the statements and supporting evidence are messy, and comparisons do not appear to be "apples-to-apples". Can you make the statements more precise? or the example more strictly comparable?

Why LETKF? I'm not advocating an exhaustive pairwise comparison between every possible model in this domain, but if the primary comparison is with LETKF then that method might need to be introduced more. note also that EnKF can handle this case as well, so would have been a logical method to include on the Figrues 4 and 5. If you are not going to do that (and I am sympathetic we we all get asked to do too many comparisons in these conference submissions) then can you make the justification explicit for which comparisons you wish to make?

Figure 8: The paper talks about $\nabla_x \log P(y|x)$ and yet graph shows $\nabla_x P(y|x)$. Why?

Figure 8, figure 2: There is not color bar on the plots ot help us interpret the field values. I assume white means zero valued and the colors are bipolar? If so, I have a further question: The plots show the score function evaluated at sparse spatial locations by showing fewer pixels non-zero with values in a grid of fixed size, implying that gradient of the score function at these locations is also zero. To my mind, would be more correct to say that the score is not sampled at that location and has no value, rather than being zero. We would more conventionally show that by displaying a lower-resolution grid, i.e. with fewer, larger pixels. Moreover, we could imagine in-filling the grid, now? $\nabla_x \log P(y_2|x_1)$ will still convey influence, and thus a gradient between the sampling sites of  $x_1$ and $y_2$ if they are spatially near each other. Or: Am I missing something? Do I not understand the conditioning operation correctly in the score function?

---

> ### Author Response · Authors · 2024-11-18
> **Rebuttal Reviewer 3 (Part 1 of 2)**
>
> We thank the reviewer for their detailed feedback and questions! Below we have provided some additional details and explanations for the reviewer's points, in addition to the main rebuttal text above:
>
> # KLD Term
> Thanks for bringing this up, refer to main rebuttal post.
>
> # Sparseness for EnSF updates
> We'd like to note that the comparison between the Full EnSF (observe the full state) and the Latent-EnSF achieves fairly similar results, so it’s not necessarily better in every aspect. It's mainly when it comes into the assimilation problems with sparse observations where the EnSF struggles. The Latent-EnSF is able to make use of information from the training data to inform the distribution of the state given the observations, but the standard EnSF is unable to do that with sparse observations. As a result of learning from the training data, the reconstruction from the VAE could also be regularizing for some of the noise terms which are otherwise present during standard EnSF processes. Our method addresses specifically that in the latent space, there is no vanishing gradient problem due to the encoded observations and ensemble states being in the same latent space with the same reduced dimension.
>
> # Training Distribution
> We refer this point to the main rebuttal section in "Clarification on the Initial States for the Shallow Water Equations", as we were unclear in our original explanation of the system. Thanks so much for asking about this! (This part should also address the clarification on what the perturbed state means). As for the part about generalization, the standard EnSF is training-free, meaning that if we start off with a very different distribution, then as long as we are given the observations it performs accordingly. However, the Latent-EnSF leverages previous data when training the VAE to make more informed updates. As a result, it can still at least handle initial conditions which come from the same family as that which it is trained on. All of our results on the shallow water equations are run on a held out test dataset with different initial conditions than the ones from the training set, and ERA5 is trained on contiguous years and then tested on subsequent ones.
>
> # Figure 2 Details
> Figure 2 is currently just showing the sparsity of the states at each time step, and not the reconstruction. The reconstruction is still dense (see the EnSF column of Figure 6 for a sample at a single time step for the reconstruction given by EnSF at 225x sparsity, or Figure 11 for multiple time steps). But as the reviewer noted, we did not describe $\eta$ in our manuscript. $\eta$ is the height of the water in the shallow water equations, we will note that in the manuscript revision. Thanks so much for pointing this out!
>
> # Sparsity Patterns
> We refer the reviewer to the "Grid Observations and Extreme Cases" section of the main rebuttal article.

---

> > ### Comment · Reviewer_GvWu · 2024-11-22
> > **Thanks, this helps but can we drill down on the question on why the EnKF struggles with the sprase observations**
> >
> > >We'd like to note that the comparison between the Full EnSF (observe the full state) and the Latent-EnSF achieves fairly similar results, so it’s not necessarily better in every aspect. It's mainly when it comes into the assimilation problems with sparse observations where the EnSF struggles. The Latent-EnSF is able to make use of information from the training data to inform the distribution of the state given the observations, but the standard EnSF is unable to do that with sparse observations. As a result of learning from the training data, the reconstruction from the VAE could also be regularizing for some of the noise terms which are otherwise present during standard EnSF processes. Our method addresses specifically that in the latent space, there is no vanishing gradient problem due to the encoded observations and ensemble states being in the same latent space with the same reduced dimension.
> >
> > I think the authors for their engagement in this review process. I appreciate the clarifications that the authors have made. For me there is still a key sticking point which prevents me from revising my score, which is that, in revising this work we still have not produced a coherent explanation of why EnSF doesn't do well under sparse observations. To be clear, I believe that the latent encoding the authors use does in fact encode helpful smoothness in the solutions, but it is not clear to me why EnSF does not. I think explaining mathematically what is going on here empirically is necessary for this work, or we don't really understand what we have done. The score function used in vanilla EnSF can also encode smoothness in the latents, and in fact it looks to me like it does. Consider equations (8)
> >
> > $$
> > \begin{aligned}
> > \nabla_x \log P\left(x_{t, \tau} \mid y_{1: t-1}\right) & =\nabla_x \log \int P\left(x_{t, \tau} \mid x_t\right) P\left(x_t \mid y_{1: t-1}\right) d x_t \\
> > & =\int-\frac{x_{t, \tau}-\alpha_\tau x_t}{\beta_\tau^2} \omega\left(x_{t, \tau}, x_t\right) P\left(x_t \mid y_{1: t-1}\right) d x_t
> > \end{aligned}
> > $$
> >
> >  and (10)
> >
> > $$
> > \nabla_x \log P\left(x_{t, \tau} \mid y_{1: t}\right)=\nabla_x \log P\left(x_{t, \tau} \mid y_{1: t-1}\right)+h(\tau) \nabla_x \log P\left(y_t \mid x_{t, \tau}\right)
> > $$
> > Naively these both seem "dense", i.e. everywhere defined, on $x_{t,\tau}$ even if $y$ is some sparse subsampling. Section 2.3 gives us something of an explanation of what might go wrong:
> > > For example, suppose that the observation map $H\left(x_t\right)=x_t[S]$, which only make observations of $x_t$ in the dimensions from the subset $S \subset\{1, \ldots, d\}$, where $S$ may have a much smaller cardinality $|S| \ll d$. In this case, the gradient of the $\log$-likelihood $\nabla_x \log P\left(y_t \mid x_t\right)$ vanishes in the dimensions in $\{1, \ldots, d\} / S$,
> >
> > And this is the part I don't understand. If we look at $f$ as defined in (7), the gradient should not, as far as i can tell, vanish in general, because nothing constrains $f$ to be applied "pointwise", unless I am missing something. Indeed in the weather forecasting examples, we do not expect the governing equations to be pointwise; rather they encode spatial correlations, so the gradient with respect to $x$ would in general be perturbed by $y_t$ no matter how sparse it is. What am I missing? Why does it count as zero here? as far as I can tell it has not been addressed in the rebuttals thus far.
> > I would like to resolve this as I think it deeply effects the interpretation of the algorithm, and the significance of the results (e.g. if EnSF did not actually have a difficulty here when correctly implemented then it woudl reduce the importance of the claim to have "fixed" it. Clarity on this point would be necessary for the paper to be publishable IMO.

---

> ### Author Response · Authors · 2024-11-18
> **Rebuttal Reviewer 3 (Part 2 of 2)**
>
> # 4.1 Clarifications
> Here, both Latent-LETKF and Latent-EnKF are benchmarks; it is just that Latent-LETKF is the stronger one. We are stating our method (Latent-EnSF) is superior to the stronger baseline (Latent-LETKF) which is better than the weaker baseline (Latent-EnKF).
>
> As for the neural network training time, this process does add some time to the process (training depends on many factors such as epochs/architecture/data volume), but the entire process can be done offline. In problems where data assimilation is utilized, the importance of the inference time precedes the importance of the training time, as inference time efficiency is crucial for real-time applications such as weather prediction and digital twins. Our results on the synthetic dataset were evaluated on a held-out set of initial conditions (though the initial conditions come from a similar family) and ERA5 experiments were run on a held-out year, so it seems to generalize empirically.
>
> # The Purpose of LETKF
> We originally intended on including EnKF in the main text as well, but we did not want to crowd a single plot. As a result, the corresponding results are actually shown in Figure 10 in the appendix. The LETKF is a variant of the EnKF with additional enhancements; as a result, we found it sufficient to just introduce the EnKF and posit the LETKF as a superior extension. We added comparison to LETKF, see above in the general response.
>
> # Figure 8
> The reviewer is absolutely correct with the labeling, we will fix the labels in revisions. For Figure 8, we wanted to emphasize the absence of gradients in the corresponding locations which are not observed and also accentuate the degree of sparsity so we went with masked (In case of Figure 2 and 8 middle) points instead. This also helps avoid confusion with just a coarser grid, because there is no aggregation and the dynamics are still on in the original dimension. It is just that we are taking subsamples for the points.
> As for in-filling the grid, though the proposed method should work for sufficiently smooth examples, interpolation leaves out a lot of detail when applied to finely detailed problems such as weather data. As a result, we chose to have a neural network learn the mapping instead of relying on interpolation which could mess up calculations when propagated through the encoder.

---

> ### Author Response · Authors · 2024-11-22
> **Reviewer 3 Follow Up (Part 1 of 2)**
>
> We thank the reviewer for their continued engagement with the discussion and excellent questions. Below, we have provided a more mathematical formulation for the likelihood vanishing gradient problem, along with some new analysis of why this matters for the EnSF, supported by some additional experimental results to corroborate our theoretical findings.
>
> # Gradients of the Score Clarification
> We refer to the gradient vanishing problem as part of the log likelihood term of the loss only with $P(y_t | X_{t, \tau})$. As the reviewer mentions, the method is still densely supported on the prior term of $P(x_{t, \tau} | y_{1:t-1})$, but specifically in our approach we are addressing the likelihood term.
>
> Just a more mathematically rigorous explanation of the vanishing gradient problem for the likelihood, term, we note that the observation function is defined $y_t = H(x_t) + \gamma_t$. For a Gaussian noise assumption, the likelihood of the observation is $$P(y_t | x_t) \propto \exp(-\frac{1}{2}||y_t - H(x_t)||_{\Gamma_t^{-1}}^2).$$
>
> Then the score function is
>
> $$\nabla_x \log P(y_t | x_t) = (y_t - H(x_t)) \Gamma_t^{-1} H'(x_t).$$
>
> For sparse observations $H(x) = x[S]$ (from the notation in the paper), we can see that $H'(x_t)[(1...d)/S]$ is $0$.
>
> About the reviewer's note for $f$, that term is just the drift term and its specific formulation does not affect the theory of the EnSF too much other than simplifying the mathematical distribution for the SDE.
>
> Here, the simplification of the joint data assimilation problem of modeling the distribution $P(x_{t} | y_{1:t})$ into the Bayesian Filtering problem where Markovian properties apply: $P(y_t | x_t) P(x_{t} | y_{1:t-1})$. The physical model (which can take care of spatial correlations) is not connected at all with the likelihood part of the update. As a result, if the likelihood component does not take into account local correlations (by means of covariance matrices such as the EnKF), then spatial correlation within that term specifically is lost otherwise.
>
> When the observations are sparse, due to the missing gradient entries in the likelihood term, the total $L2$ norm of the gradient of the likelihood is now scaled down by a factor of $\sqrt{d/k}$ (state is $d$-dimensional, observations are $k$-dimensional). As a result, the prior term dominates by the same factor as compared to full observations. Although this doesn't detract theoretically from the score filter approach given infinitesimally small time steps, in order for the EnSF formulation to be accurate in a sparse context, it then requires that the estimated prior term $P(x_{t} | y_{1:t-1})$ is somewhat representative of the true $P(x_t)$. There are now two sources of error here:
>
> Firstly, for problems where the initial state is misspecified, the assimilation portion of the original EnSF relies heavily on the likelihood component of the score in the beginning, which can lead to very slow assimilation progress due to the total likelihood gradient norm. With observation data that is also somewhat sparse in time, the assimilation progress may not be fast enough to correct for the error which is propagated across time steps.
>
> Secondly, the prior score term is primarily estimated by Monte Carlo samples. When we have full observations, the Monte Carlo error of the prior is negligible compared to the correction by the likelihood term. However, when the observations are sparse, then due to the shrinking magnitude of the likelihood term gradient, estimation errors within the prior term are magnified. As a result, to achieve good assimilation results, the number of ensemble members also increases greatly, which defeats the purpose of the EnSF being fast and scalable for data assimilation.
>
> # Experimentation for Above Claims
> In Figure 8R, we show the relative RMSE for the standard EnSF with different ensemble sizes. We assimilate the observation data at each time step (dense in time) as compared to the other approaches which are more sparse in time (every 20 time steps). Note that for small ensemble sizes (5 and 20), the estimation error actually overrides the assimilation progress, causing the error to blow up (error source 2). For high ensemble sizes (500 and 1000) the model does fairly well, but assimilation progress per time step is still remarkably slow (error source 1), even though we assimilate 20 times more than we did for the VAE-based baselines. Therefore, the EnSF needs multiple factors to work with sparse observations: observations dense in time, large ensemble size, and an accurate model, which may not exist in real-world applications.

---

> > ### Comment · Reviewer_GvWu · 2024-11-25
> >
> > Oops! I'm sorry, I wrote $f$ instead of $P(y|x)$; that made my question harder to answer than necessary. I credit the authors for answering the correct question despite my mistake. Well done.
> >
> > OK, I believe I understand the distinction being drawn here; I had misunderstood the authors; intent with plotting the sparse likelihood observations. I now understand the purpose of the sparse gradient graph, although it seems to me to be a red herring.
> > If I have understood the authors' explanation correctly, the issue here is partly statistical (there is not much information in the update) and partly numerical (it is expensive to calculate an update which uses the information optimally because of large number of Monte Carlo updates, or expensive jacobians etc)
> >
> > If we zoom out for a moment and regard the various algorithms in the paper all as filtering problems, and we consider ther difficulties as a filtering problem, with a well defined Bayesian posterior over states to be calculated, I think we mostly need to explain the performance of the method in terms of the numerical challenges. While the EnSF "sees"a sparse gradient, the posterior likelihood $P\left(x_t \mid y_{1: t}\right)$ is the same in EnSF, Latent-EnSF and for that matter EnKF etc; if any of these algorithms perform worse, it might be 1) because it is not able to make efficient statistical use of the the observations $y_{1:t}$, and some of the likelihood update is not incorporated in the calculate of the posterior, or it might be that 2) it is statistically efficient but numerically constrained and permitting enough compute to converge to that posterior is infeasible.
> >
> > For what it is worth, I would be more comfortable with this paper if the relative performance of all of these methods was justified in those terms from the beginning. IMO $P\left(x_t \mid y_{1: t}\right)$  is the true target here, not whatever approximation EnSF managed to make of it; we don't really care what shape the observation likelihood is if it updates the prior to  the same posterior likelihood.
> >
> > I think the authors responses have taken some steps in this direction. I would be curious to hear if and how they would plan to incorporate this into the text.
> >
> > >Firstly, for problems where the initial state is misspecified, the assimilation portion of the original EnSF relies heavily on the likelihood component of the score in the beginning[…]
> > >
> > >Secondly, the prior score term is primarily estimated by Monte Carlo samples. When we have full observations, the Monte Carlo error of the prior is negligible compared to the correction by the likelihood term. However, when the observations are sparse, then due to the shrinking magnitude of the likelihood term gradient, estimation errors within the prior term are magnified. As a result, to achieve good assimilation results, the number of ensemble members also increases greatly, which defeats the purpose of the EnSF being fast and scalable for data assimilation.
> >
> > OK, it sounds like the ultimate trade-off here is computational. We interpret the VAE as being a quick means of doing something *like* an amortised smoother, which imposes smoothness of the updates upon the filter, without, for example, imposing them via a more elaborate and computationally tedious update incorporating the propagation of score function updates over the graph. Is this in the author's opinion a valid summary of our shared understanding?

---

> ### Author Response · Authors · 2024-11-22
> **Reviewer 3 Follow Up (Part 2 of 2)**
>
> # Alternative Trials and Literature Review
> We have also tried to solve for the sparsity problem of the EnSF by converting it into a smoothing problem. This is done by optimizing for the modified problem of solving for alternative density $P(x_{t} | y_{1:t+1}) \propto  P(y_{t+1} | x_{t}) P(x_{t} | y_{1:t})$. We obtain the corresponding score function by differentiating through the model using automatic differentiation to obtain the gradients. Though this works slightly better (but not by much) for sparse observations as it's able to use the spatial correlation by the propagation of the physical model, it also comes at significant additional computational cost of differentiating through the model, which can quickly become intractable for large models.
>
> Finally, just as a literature review, every other EnSF-based paper [1-4] has used full or nearly full observations and avoided sparse observations due to the vanishing gradient of the score of the likelihood.
>
> [1] Bao, Feng, Zezhong Zhang, and Guannan Zhang. "A score-based filter for nonlinear data assimilation." Journal of Computational Physics (2024): 113207.
>
> [2] Bao, Feng, Zezhong Zhang, and Guannan Zhang. "An Ensemble Score Filter for Tracking High-Dimensional Nonlinear Dynamical Systems." arXiv preprint arXiv:2309.00983 (2023).
>
> [3] Bao, Feng, et al. "Nonlinear ensemble filtering with diffusion models: Application to the surface quasi-geostrophic dynamics." arXiv preprint arXiv:2404.00844 (2024).
>
> [4] Yin, Junqi, et al. "A Scalable Real-Time Data Assimilation Framework for Predicting Turbulent Atmosphere Dynamics." arXiv preprint arXiv:2407.12168 (2024).

---

> > ### Author Response · Authors · 2024-11-24
> >
> > Dear Reviewer GvWu,
> >
> > Thank you for your time and effort in reviewing our paper and helping us polish it with your insightful feedback and follow-up question. We hope that our new response has addressed your remaining concerns. Please don't hesitate ask if you have any further questions on the topic above!
> >
> > Thank you very much!
> >
> > Sincerely,
> > Authors

---

> ### Author Response · Authors · 2024-11-25
>
> Thank you for your response and your additional inputs here!
>
> > OK, I believe I understand the distinction being drawn here; I had misunderstood the authors; intent with plotting the sparse likelihood observations. I now understand the purpose of the sparse gradient graph, although it seems to me to be a red herring. If I have understood the authors' explanation correctly, the issue here is partly statistical (there is not much information in the update) and partly numerical (it is expensive to calculate an update which uses the information optimally because of large number of Monte Carlo updates, or expensive jacobians etc)
>
> We understand that the sparse gradient graph seems to be confusing without additional explanation. We will be centering the figure around explaining the lack of information in the likelihood update in our revised draft. The reviewer's understanding here is correct; the two components (statistical and numerical) combine to make the Latent-EnSF inefficient for very sparse problems. However, the numerical component is an issue which stems from the deficiency in the information from the statistical side of the issue. The problem can be so pronounced, in fact, that with our sparsity experiments in Figure 8R, we assimilate observations at the highest possible time density setting (every time step) without changing the time discretization of the simulation. Yet, even by the end of the simulation the assimilated states are still nowhere close to the target density.
>
> > If we zoom out for a moment and regard the various algorithms in the paper all as filtering problems, and we consider ther difficulties as a filtering problem, with a well defined Bayesian posterior over states to be calculated, I think we mostly need to explain the performance of the method in terms of the numerical challenges.
>
> Yes, all of these methods try to model the same true density $P(x_t | y_{1:t})$, but they have different methods of doing so through assumptions of the distribution (e.g. EnKF assuming that the density is Gaussian distributed) and methods to make computation more feasible. We go over the general formulation of all such methods in the section on Bayesian filtering, but we will edit the manuscript to further highlight the commonalities before moving onto the distinct properties of each, and use the numerical and statistical components of the loss to further motivate the Latent-EnSF before introducing our method. In particular, the Latent-EnSF addresses the information loss by supplementing it with learned generalizations from the training data, which by extension draws the numerical component back to equilibrium as in the vanilla EnSF with dense observations.
>
> Finally, we will be replacing Figure 2 with a further polished version of Figure 8R supported with our analysis of the specific weaknesses of the EnSF in sparse contexts.
>
> > OK, it sounds like the ultimate trade-off here is computational. We interpret the VAE as being a quick means of doing something like an amortised smoother, which imposes smoothness of the updates upon the filter, without, for example, imposing them via a more elaborate and computationally tedious update incorporating the propagation of score function updates over the graph. Is this in the author's opinion a valid summary of our shared understanding?
>
> We believe the summary is a good summary of the contributions of the paper. In addition to the summary above, the VAE not only encourages update smoothness but also incorporates the additional information in the form of the model weights (derived from the offline training process on the training dataset). This allows for a much faster assimilation process while keeping the statistical properties valid as compared to the slow decay of the assimilation error for the vanilla EnSF.

---

> > ### Comment · Reviewer_GvWu · 2024-11-26
> >
> > I think the reviewers for a productive an interesting review period. They have substantially persuaded me that not only are their empirical results good, but so are the underlying ideas.
> > The revised manuscript improves, but does not wholly fix the lack of clarity regarding the poor performance of the EnSF. Nonetheless, I now think this is an admissible publication and I have revised my score upwards accordingly.

---

> > > ### Author Response · Authors · 2024-11-26
> > >
> > > Dear Reviewer GvWu,
> > >
> > > Thank you so much for your time spent helping us polish the paper with your insightful questions! It's greatly appreciated.
> > >
> > > Sincerely, Authors

---

### Official Review · Reviewer_76az · 2024-11-03

**Soundness:** 3
**Presentation:** 3
**Contribution:** 2
**Rating:** 6
**Confidence:** 4

**Summary:**

The paper presents a novel data assimilation method called Latent-EnSF aimed at improving accuracy and efficiency in modeling high-dimensional and nonlinear systems with sparse observations. The authors propose using a coupled VAE with two encoders, one for the full state and another for sparse observations. This ensures consistent encoding of both types into a shared latent space, allowing efficient assimilation with fewer dimensions. Overall, the research topic is of interest in the community of data assimilation. However, from a methodology point of view, the novelty of the proposed approach might be limited to the combination of existing ENSF and latent data assimilation algorithm.

**Strengths:**

The Latent-EnSF method presented in this paper addresses key challenges in data assimilation for high-dimensional systems with sparse observations by leveraging a coupled VAE to compress both state and observation data into a shared latent space (However this idea is not new). This approach not only reduces computational complexity but also enhances the robustness of data assimilation when observation data is sparse, avoiding the gradient vanishing issues that hinder ENSF. Experimental results demonstrate Latent-EnSF's superior accuracy, faster convergence, and computational efficiency compared to existing methods.

**Weaknesses:**

The Latent-EnSF builds upon EnSF by integrating latent representations that avoid the computational challenges of sparse data. While the idea is sound and logical, performing data assimilation in the latent space is a well-established approach (see, for example, Peyron et al., 2021, 'Latent Space Data Assimilation Using Deep Learning'). Additionally, conducting data assimilation with a shared latent space for both the state vector and observations is also a known concept (see, for example, Cheng et al., 2024, 'Multi-domain Encoder–Decoder Neural Networks for Latent Data Assimilation in Dynamical Systems'). Therefore, from the reviewer’s perspective, the novelty of this paper primarily lies in combining existing latent data assimilation techniques with ENSF.

As the authors mentioned, since the state and observation vectors are encoded into a common latent space, the transformation function is restricted to an identity mapping. However, this may pose a problem regarding the specification of the observation error covariance matrix in the latent space; specifically, how to transform uncertainties from the full physical space to the latent space remains an important question. In the numerical experiments, the authors should also include noisy observations and demonstrate how latent-ENSF could account for observation uncertainty.

**Questions:**

A lot of latent data assimilation algorithms make use of the variational DA framework instead of enkf, for ex, Peyron et al., 2021 and Melinc et al, 2023. The authors should also compare 4d-var based latent DA approaches against latent-ENSF.

 Instead of only sparse observations, it would also be interesting to compare latent-ensf against existing methods for partial and irregular observations.

---

> ### Author Response · Authors · 2024-11-18
> **Rebuttal Reviewer 2**
>
> We thank the reviewer for their detailed feedback and questions! Below we have provided some additional details and explanations for the reviewer's points in addition to the main rebuttal section above:
>
> # Novelty Concerns
> The reviewer has a valid concern that the method is a combination of the standard EnSF and latent assimilation which already exists. However, this combination that we point out solves a key weakness of EnSFs which are much less apparent in other Bayesian filtering approaches, which is crucial if it is to be applied to real world applications. We note this drawback both experimentally and mathematically, and propose a corresponding solution to address this solution by adapting inspiration from existing methods. Additionally, in Appendix D, we note some additional modifications in rescaling we made to the standard EnSF as a direct plug-and-play did not work. We will cite the additional works that the reviewer mentioned in the revised manuscript.
>
> # Transforming Uncertainties from Full Physical Space to Latent Space
> We refer the reviewer to the "Latent Observation Noise Computation" section in the main rebuttal text.
>
> # Noisy Ablation Experiments
> We refer this point to the main rebuttal section with some ablation experiments with noise, in addition to adding error bars.
>
> # 4D-Var and Partial Observations
> We are solving a filtering problem to update the current state with new observations, which is different from a smoothing problem to update all previous states using all the observations in a long time window, as solved by 4D-Var. As a result, we only compare against other methods for filtering, which have the same amount of information flowing in at each time step.
>
> As for partial observations, most numerical systems are local in nature; as a result, the sparsity problem is a particularly hard problem to address compared to partial observations which may provide much more informative local information which can make the task easier for the simulation/surrogate model. For alternative observation styles, we refer the reviewer to the "Grid Observations and Extreme Cases" section in the main rebuttal text.

---

> > ### Author Response · Authors · 2024-11-24
> >
> > Dear Reviewer 76az,
> >
> > Thank you for your time and effort in reviewing our paper and helping us polish it with your insightful feedback. As the end of discussion period draws near, we would be grateful you could let us know you have received our responses, and if they have addressed your concerns and improved your evaluation of the paper. Please don't hesitate ask if you have any further questions.
> >
> > Thank you very much!
> >
> > Sincerely,
> > Authors

---

> > > ### Comment · Area_Chair_F8xV · 2024-11-26
> > >
> > > Dear reviewer,
> > > Please make to sure to read, at least acknowledge, and possibly further discuss the authors' responses to your comments. Update or maintain your score as you see fit.
> > > The AC.

---

> ### Comment · Reviewer_76az · 2024-11-26
>
> I thank the authors for addressing my comments and apologize for the delay in my response due to personal reasons. I believe the authors have provided an adequate response for most of my concerns in the main rebuttal letter. However, additional experiments could make the findings more convincing. Therefore, I am happy to raise my note to weak accept but I would suggest the authors considering the following point in the revised version/future work.
>
> 1.	I still believe that a comparison against variational DA would make the claim more convincing, particularly given that the experiments are conducted with partial observations of the state (control space). A simple 3D-Var or a 4d-var (although the authors mentioned that it uses an assimilation window with several observations, since M is also computed in the low-dimensional space, this should be computationally affordable) could be implemented using the state decoder  with a linear operator as the transformation operator.
>
> 2.	The authors perform the error covariance specification empirically in the latent space which is common in latent DA methods. However, the compression error itself could not be estimated in the latent space. With the advantage of low dimensionality, the authors might consider incorporating posterior covariance tuning algorithms (ref [1, 2], which were initially designed for variational DA). This challenge in latent DA is also mentioned in a recent review paper on ML+DA (ref [3]).
>
> [1] Desroziers, G., Berre, L., Chapnik, B. and Poli, P., 2005. Diagnosis of observation, background and analysis‐error statistics in observation space. Quarterly Journal of the Royal Meteorological Society
> [2] Ménard, R., 2016. Error covariance estimation methods based on analysis residuals: Theoretical foundation and convergence properties derived from simplified observation networks. Quarterly Journal of the Royal Meteorological Society
> [3] Cheng, S., Quilodrán-Casas, C., Ouala, S., Farchi, A., Liu, C., Tandeo, P. et al, 2023. Machine learning with data assimilation and uncertainty quantification for dynamical systems: a review. IEEE/CAA Journal of Automatica Sinica

---

> > ### Author Response · Authors · 2024-11-27
> >
> > We thank the reviewer for their additional feedback. We will aim to have the suggested additional revision suggestions in the manuscript by the camera-ready version since it may take some time for additional baseline implementation. Also, we many not have enough room for the compression error experiments in the main paper due to the page limit requirements, so the experimental results for point 2 will probably only be in the appendix.
> >
> > Nonetheless, the additional suggestions are extremely valued. Thanks so much for helping us polish this paper!

---

### Official Review · Reviewer_nywr · 2024-11-03

**Soundness:** 2
**Presentation:** 2
**Contribution:** 1
**Rating:** 5
**Confidence:** 3

**Summary:**

The Ensemble Kalman Filter (EnKF) enables data assimilation for Nonlinear State-Space Models, but it encounters the problem of vanishing gradients in tasks where observations are sparse in high-dimensional state spaces. This paper proposes a solution to this issue by proposing the Latent Ensemble Score Filter (Latent-EnSF), which embeds states and observations into a latent space via a Variational Autoencoder (VAE). The proposed method claims to be effective for various data assimilation tasks with sparse observations in high-dimensional state spaces.

**Strengths:**

This paper's strength lies in demonstrating that data assimilation in the latent space can effectively solve the problem of vanishing gradients.

**Weaknesses:**

One issue with this paper is that it unfairly compares the LETKF under unfavorable conditions. The Ensemble Kalman Filter does not inherently suffer from the vanishing gradient problem, so embedding into the latent space is not necessarily required. If the goal is to propose a good method for data assimilation in high-dimensional nonlinear state spaces, it should be directly compared with the Localized Ensemble Transform Kalman Filter (LETKF) but not with Latent-LETKF. However, this comparison is not provided.

Moreover, physical laws are often described by differential equations with spatially local interactions, leading to strong local correlations. The LETKF leverages this locality to scale to data assimilation problems with large state spaces by performing local computations in parallel. Comparing the computation speed without considering this parallelization is also misleading.

Regarding Equations (11)-(13), it is mentioned that the VAE's loss function minimizes the empirical loss function, but isn't the state ( $x_t$ ) unobserved? It is not clearly stated how learning is performed when the state ( $x_t$ ) is unobserved.

Additionally, it implicitly assumes the state can be estimated from observations at a single time point as shown in Equation (11) ($x_t$ can be recovered either from $D(z_t)$ or $D(z_t^{obs})$). Thus, it is unclear whether this method applies to nonlinear observations.

# LETKF is included in the experiments thanks to the authors' feedback. However, how much the initial conditions affect the performance is still unclear.

**Questions:**

What initial state distributions are used for the experiments? It should be explicitly given.
Regarding the perturbed trajectory, how is the perturbation applied? It should also be explicitly given.

---

> ### Author Response · Authors · 2024-11-18
> **Rebuttal for Reviewer 1**
>
> We thank the reviewer for their detailed feedback and questions! Below we have provided some additional details and explanations to the reviewer's points to expand upon the main rebuttal section up above:
>
> # Standard LETKF Comparisons
> The reviewer makes an excellent point for comparing a standard LETKF instead of Latent-LETKF for high dimensional linear spaces. Therefore, we have conducted additional experiments on the shallow water equations with the standard LETKF and sparse observations in the main rebuttal response. As in line with other reviewers' requests, we have also included results with differing amounts of noise as well as error bars plotted over 10 different initializations of the problem in Figures 1R-3R in the main rebuttal post. From these results, we can see that our Latent-EnSF achieves assimilation accuracy higher than that of LETKF, and the latent encoding inherently helps speed up the process of the assimilation, which gets tougher for the standard LETKF as the amount of noise increases.
>
> # Parallelization Speed
> The reviewer has also mentioned adding the parallelization speed for the Latent-LETKF. Doing parallelization tasks would require us to overhaul the baseline codebase for LETKF quite a bit, but we can establish a lower bound for the computational time of the Latent-LETKF approach by just dividing the wall clock time by a factor of the number of cores (which for the purposes of this let's say is 64). For the numbers we provide in the table, this would be 0.103 for the shallow water equations, and 7.035 for the ERA5 experiments, which is in-line with our claims.
>
> # Unobserved State
> The full state $x_t$ is not used/unobserved at test time. However, since we are training the VAE model offline (see Training Process of Latent-EnSF part in the main rebuttal section), we do have the full state at train time.
>
> # Observation and State Assumptions
> Our current architecture does assume that the observations at a certain time step can be used to get a sufficient approximation of the true latent state. In a way, the observation encoder can be viewed as a noised form of the state encoder, where the noise would be regularized by the data assimilation techniques in the latent space. As a result, there is a certain degree of sparsity that the model is able to handle. However, as demonstrated by our experiments, we can afford quite a high degree of sparsity and information loss with a 225x sparsity reduction on the shallow water equation experiments. For sparsity of such degree, other standard data assimilation methods would also have a particularly difficult time conducting accurate data assimilation, as shown by our results with the standard EnSF and LETKF.
> Additionally, alternative architectures making use of past information such as LSTM-based encoders would resolve some of the additional issues.
>
> # Clarification on the Initial States for the Shallow Water Equations
> We refer this point to the main rebuttal section, as we were unclear in our original explanation of the system. Thanks so much for asking about this!

---

> > ### Author Response · Authors · 2024-11-24
> >
> > Dear Reviewer nywr,
> >
> > Thank you for your time and effort in reviewing our paper and helping us polish it with your insightful feedback. As the end of discussion period draws near, we would be grateful you could let us know you have received our responses, and if they have addressed your concerns and improved your evaluation of the paper. Please don't hesitate ask if you have any further questions.
> >
> > Thank you very much!
> >
> > Sincerely,
> > Authors

---

> > ### Comment · Reviewer_nywr · 2024-11-25
> > **Behavior of the data assimilation and Sensitivities on initial states**
> >
> > Thanks for responding to my comments.
> > I greatly appreciate the authors' responses but do not fully understand the results.
> > Let me ask some questions about the experiments.
> >
> > For SWE experiments, the authors include the results of LETKF in Figures 1R to 4R.
> > As can be seen from the figures, the behaviors of latent-LETKF,  latent-EnSF, and latent-EnKF are consistent overall, rapidly reducing the RMSE and gradually increasing the RMSE as time passes.
> > On the other hand, LETKF gradually decreases the RMSE as time passes.
> > What is the reason on this behaviors?
> > It looks LETKF gradually improves the state guess as the observation increases.
> > Does it imply the initial guess of LETKF is terrible compared with the other methods?
> >
> > Regarding the initial states, how is the sensitivities on the initial states?
> > Could you show or explain the dependence of the performance on the initial guess of the states?

---

> ### Author Response · Authors · 2024-11-25
>
> We thank the reviewer for their response and follow up questions!
>
> The initial guess for all methods are about the same; they start at exactly the same set of misspecified perturbed states. The latent methods assimilate much faster due to the additional generalization coming from the machine learning model (also see the Vanilla EnSF analysis chart in Figure 8R where there is also information loss coming from the sparse observations). In fact, you can see that with different noise levels, the LETKF actually assimilates at different speeds depending on the amount of noise injected.
>
> The phenomenon for the error increasing for the latent methods is primarily due to the dynamics becoming more complex as time goes on due to wave interference patterns, which causes slightly more inaccuracies in the reconstruction. The latent methods assimilate much faster due to the additional generalization coming from the neural network model. We'd like to emphasize that as the amount of observation noise grows, LETKF seems to have an increasingly difficult time assimilating the noise compared to the latent methods.
>
> Additionally, note that the error increasing is much more apparent due to the logarithmic scale we chose to plot the results for the rebuttal in. The actual discrepancy by the end is quite wide without the log transformation.
>
> For the different initial states, the perturbation is with the state parameters. We found that any initial state perturbation of above 5% in the parameters results in a completely off estimate (Relative RMSE > 1) by the last time step. In our experiments, we adopt a parameter perturbation of about 10%. As a result, the initial estimate does not play any significant role in our experiments up above.

---

> > ### Comment · Reviewer_nywr · 2024-12-01
> > **Comparison with the other methods**
> >
> > > For the different initial states, the perturbation is with the state parameters. We found that any initial state perturbation of above 5% in the parameters results in a completely off estimate (Relative RMSE > 1) by the last time step.
> >
> > Let me clarify the sensitivities regarding the initial guess. How do the other methods behave under different initial state perturbations? Did all the methods fail to estimate the state when the initial perturbation is above 5%? The authors claim that their proposed method assimilates much faster due to the additional generalization. However, does this only hold when the initial guess is close to the true state? Based on the current experimental results, it is still difficult for me to judge whether the authors' statements are fair and justified.

---

> > > ### Author Response · Authors · 2024-12-01
> > >
> > > We thank the reviewer for their continued discussion. We would like to clarify the primary misunderstanding with the reviewer regarding the question below which should hopefully clear up the remaining questions:
> > >
> > > > However, does this only hold when the initial guess is close to the true state?
> > >
> > > The initial statement of the initial perturbations is with regards to the performance of the ground truth model without data assimilation (perturbed line for Figure 3 and Figures 1R-3R). Our statement quoted by the reviewer is just to demonstrate that the initial state is sufficiently different from the ground truth state, or in other words when the initial guess is far from the true state.
> > >
> > > All data assimilation methods are performed and evaluated with respect to a 10% perturbed state which originally would fail without any data assimilation, but these methods are able to improve upon the performance to varying degrees. If the reviewer is worried that the initial perturbed state still technically comes from a similar family of initial conditions, we achieved similar results when we considered the initial state estimate for all assimilation methods to just be zeroes across the board, so all the methods are very robust to the initial condition.

---

> > > > ### Author Response · Authors · 2024-12-01
> > > >
> > > > In case the reviewer wanted to also check out the results for "we achieved similar results when we considered the initial state estimate for all assimilation methods to just be zeroes across the board, so all the methods are very robust to the initial condition", we also ran the full suite again (except for the vanilla LETKF due to time constraints) for this setting and have placed it in Figure 9R-11R, which shows similar results to the perturbed experiments.

---

### Author Response · Authors · 2024-11-18
**Main Rebuttal (Part 1 of 2)**

We thank the reviewers for your detailed feedback and constructive suggestions. In this main section of the rebuttal, we will tackle some common points by the reviewers. All experimental results will be using this link: https://imgur.com/a/xJhxSrn. Figures using this album will be referred to with an R after the number, e.g. Figure 1R, as contrasted with standard figures in the paper.

# Robustness of Experimental Results, and Comparison against a Standard LETKF Baseline
Above, we show experimental results on the shallow water equation setup. For this current comparison, we compare the results for the 400-dimensional latent methods, along with the LETKF in the original/full space. We additionally include noise ablation experiments corresponding to no noise, 10\% noise and 20\% noise for each of the plots (shown in Figures 1R-3R). The results also include error bars given by the standard deviation across 10 different random initial conditions and perturbations. This set of experiments demonstrates the robustness of our method and its advantage over LETKF. We will add the full version of all the plots in the manuscript revision at a later date.

# Training process of Latent-EnSF
The VAE is only trained offline using  offline data. During online data assimilation, only forward passes of VAE are needed for inference.

# Latent Matching
Intuitively, the latent representation of the observation encoder should in expectation match that of the state encoder, but with additional uncertainty due to the loss of information. We slightly revised the model architecture after the submission of the paper, which achieved improved VAE reconstruction accuracy by matching only means in the current representation of the problem (artifact from earlier version, changes will be reflected in the revised version of the paper).

# KLD Term
The reviewers have brought up that the KLD term is not the actual KL-Divergence; it is indeed the KL-Divergence from a diagonal Gaussian to the standard Gaussian distribution, which is shown/implied from equation (14) but not explicitly stated. We will add this detail in the revised version of the paper, and thanks so much for pointing it out!

However, this part is not very important, as our focus is on the quality of the latent representation, rather than the generative model properties. In future iterations of the experiments, we found that using a variant of the autoencoder similar to what’s used in latent diffusion models [1] worked better, and those have much less KLD weighting, which we multiplied by 1e-5.

# Latent Observation Noise Computation
Some reviewers mentioned that direct transformation of uncertainties from the full physical space to the latent space can be quite tricky, and this is true for multiple reasons. Firstly, the uncertainties are propagated by the nonlinearities given by the encoder network. Secondly, there is a loss of information coming from the sparse observations, resulting in more uncertainty. These are hard to directly convert from full physical space to latent space.

For our experiments, we encode (by means of the observation encoder) noisy observations from the full space and compare it against the true encoded state (given by the state encoder) at train time. This allows us to get a value in the neighborhood of the true uncertainty in the latent space. Note that the uncertainty does not need to be exact; covariance inflation techniques are commonly applied to EnKF methods to make them more robust to possible uncertainties which may not be well defined in the state space either.

We additionally generated noise plots for our shallow water experiments where we calculate the distribution of the latent error between the true latent state (given by encoding the non-noised state) and the latent encoding of the noisy observations, for two different levels of noise, which we show in Figure 6R and Figure 7R. We measure these for 10 different trajectories. Note that the standard deviation of the latent noise is stable across all time steps, and the mean is approximately zero.

---

> ### Author Response · Authors · 2024-11-18
> **Main Rebuttal (Part 2 of 2)**
>
> # Grid Observations and Extreme Cases
> The reviewers have pointed out that we are currently testing with regards to observations on an even/uniform grid. This is a byproduct of the choice of architecture specifically, which we choose to be a standard convolutional variational autoencoder for simplicity of testing. Other autoencoder architectures such as masked autoencoders and conditional neural fields such as the one used in the ConFiLD paper can effectively take observations which are defined on non-standard grids.
>
> The reviewers also bring up another important point: what if the observations at one time step cannot be used to recover the true state? Firstly, we leverage the intrinsic low-dimensionality of the true state which can be sufficiently captured by sparse observations. If the observations are extremely sparse, it is indeed possible that the observations at one time step cannot be used to recover the true state well by the latent representation. This is a common challenge for other models to assimilate as well. In this case, by incorporating temporally-based architectures within the VAE such as LSTMs or the analogous part in ConFiLD, this problem can be potentially mitigated. We will add these comments to the revision.
>
> Finally, according to the analysis in the section above, sparsity and noise both introduce uncertainty in the model, which translates to the latent space. In Figure 4R and Figure 5R, we additionally plot the reconstruction error for the sparse observational reconstruction and the full state reconstruction in comparison to the data-assimilation methods. We note that the Latent-EnSF method, in fact, surpasses pure reconstruction due to its ability to regularize the noise that is propagated to the latent state.
>
> # Clarification on the Initial States for Shallow Water Equations
> The initial state distributions of the shallow water equation experiments that we used are parameterized by a Gaussian bump (for water height, not for $u$ or $v$ velocity) centered at a random point $x, y$ in the $L$ by $L$ grid, with standard deviation $L/2$. Additionally there is Gaussian noise added in the beginning.
> For the old experiments, we predefined a perturbed center for the Gaussian bump $x', y'$ and simulated the corresponding dynamics. For the updated experiments for shallow-water equations, we add random perturbation $x', y' = x'+U(0, 0.1L), y'+U(0, 0.1L)$. Note that these are not just random noise, they're changes to the location of the initial wave, which can lead to massive errors early in simulation.
> We note that we have not been clear in explaining the experimental setup, and will revise our manuscript to include the specific details.
>
> [1] Robin Rombach and Andreas Blattmann and Dominik Lorenz and Patrick Esser and Björn Ommer, High-Resolution Image Synthesis with Latent Diffusion Models, 2021.

---

### Comment · Area_Chair_F8xV · 2024-11-26

Dear all,

The deadline for the authors-reviewers phase is approaching (December 2).

@For reviewers, please read, acknowledge and possibly further discuss the authors' responses to your comments. While decisions do not need to be made at this stage, please make sure to reevaluate your score in light of the authors' responses and of the discussion.

- You can increase your score if you feel that the authors have addressed your concerns and the paper is now stronger.
- You can decrease your score if you have new concerns that have not been addressed by the authors.
- You can keep your score if you feel that the authors have not addressed your concerns or that remaining concerns are critical.

Importantly, you are not expected to update your score. Nevertheless, to reach fair and informed decisions, you should make sure that your score reflects the quality of the paper as you see it now. Your review (either positive or negative) should be based on factual arguments rather than opinions. In particular, if the authors have successfully answered most of your initial concerns, your score should reflect this, as it otherwise means that your initial score was not entirely grounded by the arguments you provided in your review. Ponder whether the paper makes valuable scientific contributions from which the ICLR community could benefit, over subjective preferences or unreasonable expectations.

@For authors, please respond to remaining concerns and questions raised by the reviewers. Make sure to provide short and clear answers. If needed, you can also update the PDF of the paper to reflect changes in the text. Please note however that reviewers are not expected to re-review the paper, so your response should ideally be self-contained.

The AC.

---

### Meta-Review · Area_Chair_F8xV · 2024-12-20

**Metareview:**

The reviewers recommend acceptance (5-6-6-8-6-6). The paper proposes Latent Ensemble Score Filter for improving data assimilation in high-dimensional and nonlinear systems. The approach is sound and the results are good. The author-reviewer discussion has been very constructive and has led to a number of clarifications and improvements that have convinced the reviewers. Given the overall positive reviews, I recommend acceptance. I request the authors to implement the many suggestions made during the author-reviewer discussion period in the final version of the paper.

**Additional Comments On Reviewer Discussion:**

The author-reviewer discussion has been very constructive and has led to a number of clarifications and improvements that have convinced the reviewers.

---

### Decision · Program_Chairs · 2025-01-22

Accept (Poster)